# SymILO: A Symmetry-Aware Learning Framework for Integer Linear Optimization

Qian Chen[1,2], Tianjian Zhang[1,2], Linxin Yang[2,3], Qingyu Han[2], Akang Wang[2,3,*], Ruoyu Sun[3,4], Xiaodong Luo[2,3], and Tsung-Hui Chang[1,2]

[1]School of Science and Engineering, The Chinese University of Hong Kong, Shenzhen, China
[2]Shenzhen Research Institute of Big Data, China
[3]School of Data Science, The Chinese University of Hong Kong, Shenzhen, China
[4]Shenzhen International Center For Industrial And Applied Mathematics, Shenzhen Research Institute of Big Data, China

## Abstract

Integer linear programs (ILPs) are commonly employed to model diverse practical problems such as scheduling and planning. Recently, machine learning techniques have been utilized to solve ILPs. A straightforward idea is to train a model via supervised learning, with an ILP as the input and an optimal solution as the label. An ILP is symmetric if its variables can be permuted without changing the problem structure, resulting in numerous equivalent and optimal solutions. Randomly selecting an optimal solution as the label can introduce variability in the training data, which may hinder the model from learning stable patterns. In this work, we incorporate the intrinsic symmetry of ILPs and propose a novel training framework called SymILO. Specifically, we modify the learning task by introducing solution permutation along with neural network weights as learnable parameters and then design an alternating algorithm to jointly optimize the loss function. We conduct extensive experiments on ILPs involving different symmetries and the computational results demonstrate that our symmetry-aware approach significantly outperforms three existing methods—-achieving $50.3\%$, $66.5\%$, and $45.4\%$ average improvements, respectively.

## 1 Introduction

Integer linear programs (ILPs) are optimization problems with integer variables and a linear objective, and have a wide range of practical uses in various fields, such as production planning (Pochet & Wolsey, 2006; Chen, 2010), resource allocation (Liu & Fan, 2018; Watson & Woodruff, 2011), and transportation management (Luathep et al., 2011; Schöbel, 2001). An important property that often arises in ILPs is *symmetry* (Margot, 2003), which refers to a situation where permuting variables does not change the structure of an ILP.

Recently, there emerges many approaches equipping machine learning methods, supervised learning in particular, to help efficient solution identification for ILPs (Zhang et al., 2023). Among these approaches, an important category derived from the idea of predicting the optimal solution has demonstrated significant improvements (Han et al., 2023; Ding et al., 2020; Khalil et al., 2022; Nair et al., 2020). In this paper, we consider a classic supervised learning task that aims to train an ML model to predict an optimal solution for an ILP. Specifically, given a training dataset $\mathcal{D} = \{(s_i, y_i)\}_{i=1}^N$ with $y_i$ denoting an optimal solution to instance $s_i$, we hope to train a neural network

---

*Corresponding author: Akang Wang <wangakang@sribd.cn>

38th Conference on Neural Information Processing Systems (NeurIPS 2024).

model $f_\theta(\cdot)$ to approximate the mapping from ILP instances to their optimal solutions, via minimizing the empirical risk defined on $f_\theta(s_i)$ and $y_i$.

However, for an ILP $s_i$ with symmetry, there exist multiple optimal solutions including $y_i$ and its symmetric counterparts, any of which has an equal probability of being returned as a label. Training neural networks without taking symmetry into account is basically learning a model supervised by random outputs, leading to prediction models of inferior performance.

To address this issue, we propose to leverage the symmetry of ILPs to improve the model performance of predicting an optimal solution. Specifically, given input $s_i$, we define a new empirical risk using $f_\theta(s_i)$ and $\pi_i(y_i)$, where $\pi_i(\cdot)$ denotes the operation of permuting elements in $y_i$ into its symmetric counterpart. Along with ML model parameters, the permutation operators will also be optimized during training. To achieve this, we further develop a computationally affordable algorithm that alternates between optimization of model parameters and optimization of permutation operation. The distinct contributions of our work can be summarized as follows.

- We propose a symmetry-aware framework (called SymILO) that introduces permutation operators as extra optimization variables to the classic training procedure.

- We devise an alternating algorithm to solve the newly proposed problem, with a specific focus on updating the permutation operator for different symmetries.

- We conduct comprehensive numerical studies on four typical benchmark datasets involving symmetries, and the results show that our proposed approach significantly outperforms existing methods.

## 2 Related works

Previous works on identifying high-quality solutions to ILPs via machine learning techniques mainly focus on reducing problem sizes. For example, Ding et al. (2020) propose to identify and predict a subset of decision variables that stay unchanged within the collected solutions. Li & Wu (2022) formulate MILPs as Markov decision processes and learn to reduce problem sizes via early-fixing.

It is noteworthy that the emergence of GNNs has had a significant impact on solving ILPs. Gasse et al. (2019) are the first to propose a bipartite-graph representation of ILPs and pass it to GNNs. Nair et al. (2020) adopt the same representation scheme and train GNNs to predict the conditional distribution of solutions, from which they further sample solutions. Rather than directly fixing variables, Han et al. (2023) conduct search algorithms in a neighborhood centered around an initial point generated from the predicted distribution. Other works based on GNNs (Sonnerat et al., 2022; Lin et al., 2019; Khalil et al., 2022; Wu et al., 2021) also illustrate great potential in improving the solving efficiency.

Limitations of the existing GNN-based approaches are also noticed. Nair et al. (2020); Han et al. (2023) try to address the multiple solution problem by learning the conditional distribution. Chen et al. (2022) introduce random features into the bipartite graph representation to differentiate variable nodes involving symmetries.

However, none of the existing learning-based approaches explicitly leverage the inherent symmetries in ILPs to achieve improvements. In contrast, works from mathematical optimization perspectives suggest that symmetry-handling algorithms exhibit great abilities in solving symmetry-involving ILPs (Pfetsch & Rehn, 2019). To name a few, such algorithms include orbital fixing (Ostrowski et al., 2011), tree pruning (Margot, 2002), and lexicographical ordering (Kaibel & Pfetsch, 2008).

## 3 Background and preliminaries

### 3.1 ILPs

An *integer linear program* (ILP) has a formulation as follows:

$$\min_x \ \{c^\top x | Ax \le b, x \in \mathbb{Z}^n\} \tag{1}$$

where $x \in \mathbb{Z}^n$ are integer decision variables, and $c \in \mathbb{R}^n, A \in \mathbb{R}^{m \times n}, b \in \mathbb{R}^m$ are given coefficients.

## 3.2 Symmetry Group

Symmetry of ILPs is typically represented by groups. We start with some basic notations and most of which follow Margot (2009). Denoting the index set by $I^n = \{1, 2, \ldots, n\}$, a *permutation* on $I^n$ is a bijective (one-to-one and onto) mapping $\pi : I^n \to I^n$. For example, an identity permutation maps the index set to itself as $\{\pi(i) = i\}_{i=1}^n$ and a cyclic permutation has rotational mapping rules $\{\pi(i) = i + 1\}_{i=1}^{n-1}$ and $\pi(n) = 1$. Schematic diagrams of these two permutations and other ones are shown in Figure 1. For brevity, we abuse the notation $\pi$ a little bit and denote the permutation acting on a vector $y \in \mathbb{R}^n$ by rearranging its coordinates, namely $\pi(y) = \left[ y_{\pi(1)}, y_{\pi(2)}, \ldots, y_{\pi(n)} \right]^\top$. Let $Q$ be the set of all feasible solutions of (1) and $S_n$ the set of all permutations on $I^n$. Note that $S_n$ is referred to as the **symmetric group**, which should not be confused with the **symmetry group** discussed as follows.

**Definition 3.1.** A *symmetry group* of (1) is defined as the set of all permutations $\pi$ that map $Q$ onto itself, such that each feasible solution is mapped to another feasible solution with the same objective value, i.e.,

$$G = \{\pi \in S_n : c^\top \bar{y} = c^\top \pi(\bar{y}) \text{ and } \pi(\bar{y}) \in Q, \ \forall \bar{y} \in Q\}. \tag{2}$$

Next, we will delve into three commonly encountered symmetries, accompanied by typical example problems. Since not all variables in an ILP involve symmetry, we use $q \le n$ to indicate the size of the symmetry group.

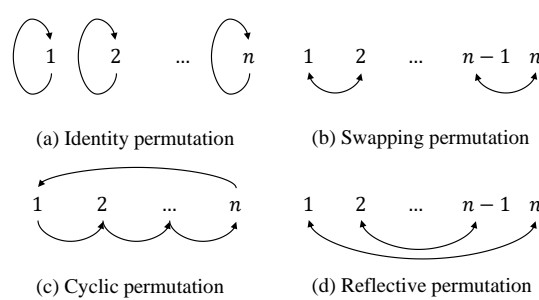

(a) Identity permutation      (b) Swapping permutation

(c) Cyclic permutation      (d) Reflective permutation

**Symmetric group** The *symmetric group*, denoted by $S_q$, is the group that consists of all permutations ($q!$ in total) on $I^q$. Problems with this kind of symmetry include bin packing (Johnson, 1974) and optimal job scheduling Graham et al. (1979), etc. An example is illustrated in Appendix B.0.1.

Figure 1: Permutation examples with directed edges denoting mapping rules.

**Cyclic and dihedral groups** As its name suggests, cyclic symmetry allows elements to be permuted to their right neighbors, cycling the right-most variables back to the left, e.g., a cyclic (or rotational) permutation $\rho$ in Figure 1 (c). The elements of a *cyclic group* $C_q$ are powers of $\rho$, and $|C_q| = q$. Problems with cyclic group often have characteristics of rotations or cycles, e.g., periodic event scheduling problem (Serafini & Ukovich, 1989).

Compared to the cyclic group, a *dihedral group* (denoted as $D_q$) additionally includes reflective permutations, which is illustrated in Figure 1 (d). Consequently, $D_q$ comprises a total of $2q$ distinct permutations. A typical problem with such symmetry is the circular (or modular) golomb ruler problem (see Appendix B.0.2).

## 3.3 Classic supervised learning for solution prediction

A classic solution prediction task based on supervised learning is formulated as follows. Let $\mathcal{S}$ be the space of ILP instances and $\mathcal{Y}$ be the label (i.e., optimal solution) space. A model function $f_\theta : \mathcal{S} \to \mathcal{Y}$ parameterized by $\theta \in \Theta$ is used to learn a mapping from instances to optimal solutions. Let $\mathcal{P}(S, Y)$ be a distribution over $\mathcal{S} \times \mathcal{Y}$. The performance of the model function is measured by a criterion called *true risk* : $R(f_\theta) := E_{\mathcal{P}(S,Y)} \left[ \ell \left( f_\theta(s), y \right) \right]$, where $\ell : \mathcal{Y} \times \mathcal{Y} \to \mathbb{R}^+$ is a given loss function, e.g., mean squared error or cross-entropy loss. An intuitive way to improve the model performance is to minimize the *true risk*. However, one cannot access all data from distribution $\mathcal{P}(S, Y)$, which makes it impossible to calculate the true risk. Practically, one can obtain a set of (instance, solution) pairs called training data $\mathcal{D} = \{(s_i, y_i)\}_{i=1}^N \subseteq (\mathcal{S} \times \mathcal{Y})^N$ sampled from $\mathcal{P}(S, Y)$, based on which define the *empirical risk* as

$$r(f_\theta; \mathcal{D}) := \frac{1}{N} \sum_{i=1}^N \ell \left( f_\theta(s_i), y_i \right). \tag{3}$$

By minimizing the *empirical risk*, i.e., $\min_{\theta \in \Theta} r$, one aims to approximate the minimization of the *true risk*, under the assumption that the training data is a representative sample of the overall data distribution.

## 4 Methodology

### 4.1 Reformulation of the learning task

In Section 3.3, we introduce a classic supervised learning task for general ILPs, which aims at learning a mapping $f_\theta$ from instances to optimal solutions. In this task, a dataset $\mathcal{D} = \{(s_i, y_i)\}$ is given, and the mapping $f_\theta$ is learned by minimizing (3) with $\theta$ as decisions. However, for ILPs with symmetry, an ILP instance has multiple solutions (let $Y_i$ be the set of optimal solutions of $i$-th instance). As a consequence, the labels in this task have multiple choices, thus datasets choosing different optimal solutions as labels $\{\mathcal{D}' = \{(s_i, y'_i)\}_{i=1}^N, \forall y'_i \in Y_i\}$ are all valid for the learning task. Empirically, we observe that different $D'$ can lead to distinct performance, which motivates us to consider the selection of labels for ILPs with symmetry.

We reformulate the learning task as follows. Firstly, we augment dataset $\mathcal{D}$ to dataset $\mathcal{D}_s = \{(s_i, y_i, G_i)\}$, where $G_i$ is the symmetry group of $i$-th instance and $\pi_i \in G_i$. Secondly, we define the *symmetry-aware empirical risk* as

$$r_s(f_\theta, \{\pi_i\}_{i=1}^N; \mathcal{D}_s) := \frac{1}{N} \sum_{i=1}^N \ell\left(f_\theta(s_i), \pi_i(y_i)\right). \tag{4}$$

Then, the mapping $f_\theta$ is learned by minimizing the symmetry-aware risk as $\min_{\theta, \pi} r_s$ (both $\theta$ and $\pi$ as decisions). In contrast to the original task, the symmetry-aware task uses symmetry information by introducing extra decisions $\{\pi_i\}_{i=1}^N$, so as to dynamically selecting proper optimal solutions as labels. There are important differences between the symmetry-aware empirical risk and the classic one:

**Proposition 4.1.** *Let $r^*$ and $r_s^*$ be the global minimal values of $\min_\theta r$ and $\min_{\theta, \pi} r_s$, respectively. Then, the following claims hold:*

  *(i) $r_s^* \leq r^*$,*

  *(ii) $r_s^* < r^*$, if there exist $i, j \in \{1, \ldots, N\}$, such that $s_i = s_j$ and $y_i \neq y_j$.*

Claim (i) always holds since $\min_\theta r$ is a special case of $\min_{\theta, \pi} r_s$ when $\pi_1, \ldots, \pi_N$ are all *identity permutations*. Claim (ii) shows a significant advantage of $r_s$ compared to $r$. A non-rigorous proof is available in Appendix A.1.

### 4.2 An alternating minimization algorithm

The minimization of (4) is challenging due to the discrete nature of $\pi$. Motivated by the well-known block coordinate minimization algorithms (Mangasarian, 1994), we update $\theta$ and $\pi$ alternately, i.e.,

$$\{\pi_i^{k+1}\}_{i=1}^N \leftarrow \arg \min_{\pi_i \in G_i} r_s(f_{\theta^k}, \{\pi_i\}_{i=1}^N; \mathcal{D}_s), \tag{5}$$

$$\theta^{k+1} \leftarrow \arg \min_{\theta \in \Theta} r_s(f_\theta, \{\pi_i^{k+1}\}_{i=1}^N; \mathcal{D}_s). \tag{6}$$

Such an alternating mechanism divides the minimization of (4) into two sub-problems: a discrete optimization in (5) over sets $\{G_i\}_{i=1}^N$ and a classic empirical risk minimization in (6). Repeatedly solving (6) to optimal is unrealistic, thus it is more practical to update $\theta$ by several gradient steps instead.

The sub-problem in (5) is further specified as shown in Section 4.2.1, according to the symmetry structures in the ILP instances.

We summarize the proposed alternating minimization algorithm in Algorithm 1. In the main loop, $\{\pi_i\}_{i=1}^N$ are updated first (line 5), after which an inner loop (lines 6-10) is operated to update $\theta$ through a gradient-based method GD, e.g., Adam (Kingma & Ba, 2014). These two updates alternate until a preset maximum number of epochs $K$ is reached. We finally note that Algorithm 1 can be easily adapted to a mini-batch version, in which the data can be randomly sampled from $\mathcal{D}_s$.

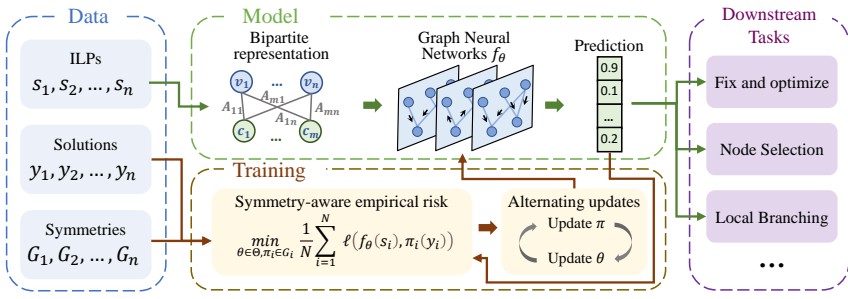

Figure 2: An overview of the SymILO framework.

### 4.2.1 Optimization over symmetry groups in (5)

In this section, we investigate the concrete formulations of the sub-problem in (5) for the symmetry groups mentioned in Section 3.2 (symmetric group, cyclic and dihedral groups), and devise algorithms to solve them.

**Cyclic and dihedral groups** The cardinality of a cyclic group $C_q$ is $q$, and it is $2q$ for a dihedral group $D_q$. For symmetry groups with such reasonably small size, a straightforward and effective way to solve (5) is to evaluate all possible permutations and select the one that yields the minimum $r_s$.

---

**Algorithm 1** Alternating optimization

1: **Input:** Dataset $\mathcal{D}_s = \{(s_i, y_i, G_i)\}_{i=1}^{N}, K, T$
2: **Output:** $\theta^K$
3: **Initialize** ML model parameter $\theta^1$;
4: **for** $k \leftarrow 1$ **to** $K - 1$ **do**
5: $\quad \{\pi_i^{k+1}\}_{i=1}^{N} \leftarrow \arg\min_{\pi_i \in G_i} r_s(f_{\theta^k}, \{\pi_i\}_{i=1}^{N}; \mathcal{D}_s)$;
6: $\quad \tilde{\theta}^1 \leftarrow \theta^k$;
7: $\quad$ **for** $t \leftarrow 1$ **to** $T - 1$ **do**
8: $\quad\quad \tilde{\theta}^{t+1} \leftarrow \text{GD}(\tilde{\theta}^t, \nabla_{\tilde{\theta}^t} r_s(\tilde{\theta}^t, \{\pi_i^{k+1}\}_{i=1}^{N}; \mathcal{D}_s))$ ;
9: $\quad$ **end for**
10: $\quad \theta^{k+1} \leftarrow \tilde{\theta}^T$;
11: **end for**

---

**Symmetric group** The cardinality of a symmetric group is factorially large, $|S_q| = q!$, so it is impractical to traverse all permutations. Since $\pi_1, \ldots, \pi_N$ are not coupled, we can separate them and solve the $N$ sub-problems individually:

$$\min_{\pi_i} \ell\left(f_\theta(s_i), \pi_i(y_i)\right), \forall i = 1, \ldots, N. \tag{7}$$

Without loss of generality, consider an ILP whose variables have a matrix form (e.g., see Appendix B.0.1), denoted by $X \in \mathbb{Z}^{p \times q}(p \cdot q < n)$, and a symmetric group $S_q$ acting on its column coordinates. In this case, (7) is equivalent to solve the following binary linear program (BLP),

$$\min_P \ell\left(\hat{X}, XP\right) \quad \text{s.t.} \quad P \in \{0,1\}^{q \times q}, P^\top \mathbf{1} = \mathbf{1}, P\mathbf{1} = \mathbf{1}, \tag{8}$$

where $P$ is a permutation matrix, $\hat{X}$ is the matrix form of $f_\theta(s_i)$, and $\mathbf{1}$ is an all-one vector. We relax $P$ to take continuous values between 0 and 1, and get a linear program (LP),

$$\min_P \ell\left(\hat{X}, XP\right) \quad \text{s.t.} \quad P \in [0,1]^{q \times q}, P^\top \mathbf{1} = \mathbf{1}, P\mathbf{1} = \mathbf{1} \tag{9}$$

According to Proposition 4.2, one can solve (9), to get the optimal permutations for the original problem in (8). It can be done quite efficiently with the aid of off-the-shelf LP solvers, such as Gurobi Optimization, LLC (2023), CPLEX IBM (2020), etc.

**Proposition 4.2.** *When $\ell$ is the squared error or binary cross-entropy loss, the optimal solution to (9) is also an optimal solution to (8). (See the proof in A.2.)*

### 4.3 An overview of the SymILO framework

In this section, we summarize a novel learning framework (SymILO) that utilizes symmetry for solving ILPs. An overview is depicted in Figure 2, which consists of two parts: the upper row connected by green arrows delineates a graph neural network (GNN)-based workflow, and the lower row connected by red arrows outlines the training process.

For the GNN-based workflow, an ILP is first converted to a bipartite graph (see appendix C for details), which is then fed to a GNN model $f_\theta$ (see appendix D for details), producing a predicted solution. Notably, the predicted solution is finally used in downstream tasks for refinement. Due to the complexity of solving ILPs, existing methods, such as Nair et al. (2020); Ding et al. (2020); Khalil et al. (2022); Han et al. (2023), often include a post-processing module taking the predicted solution as an initial point to identify higher-quality solutions. Our approach follows this routine and integrates certain downstream techniques. Section 5.1 specifies three downstream tasks.

For the training process, the data used to minimize the symmetry-aware empirical risk $r_s$ include the collected solution $y_i$ and the symmetry group $G_i$ of each instance $s_i$. Both parameters $\theta$ of the GNN model and permutations $\{\pi_i\}_{i=1}^N$ of each solution are optimized via an alternating algorithm mentioned in Algorithm 1. Given a trained model $f_{\theta^K}$, the prediction $f_{\theta^K}(s')$ for an unseen instance $s'$ is used to guide the downstream tasks in identifying feasible solutions. Note that $\{\pi_i\}_{i=1}^N$ are utilized only in the training phase but not in the inference phase.

## 5    Experimental settings

In this section, the experimental settings are presented. The corresponding source code is available at https://github.com/NetSysOpt/SymILO.

### 5.1    Downstream tasks and baselines

In our experiments, we pass the predictions of GNN models to three downstream tasks, namely fix and optimize, local branching, and node selection, to identify feasible solutions. For each downstream task, we choose one existing method as a baseline. The downstream tasks and their corresponding baselines (in parentheses) are shown below.

**Fix and optimize (ND):**    "Fix and optimize" refers to a strategy where one first "fix" or set some variables to specific values and then "optimize" the remaining variables to find better solutions. The baseline we choose is "Neural Diving" (**ND**) proposed by Nair et al. (2020), a technique using a graph neural network to generate partial assignments for ILPs, which creates smaller sub-ILPs with the unassigned variables.

**Local branching (PS):**    Local branching is a heuristic method that constructs a linear constraint based on a given initial solution to the original ILP instance. This constraint restrains the search space in a region around the initial solution. It can help guide the optimization process toward better solutions while balancing computational efficiency. Approaches based on this idea include Ding et al. (2020); Han et al. (2023); Chen et al. (2023) and we select the "predict-and-search" (**PS**) framework proposed by Han et al. (2023) as a baseline.

**Node selection (MIP-GNN):**    In branch and bound algorithms, node selection is a process of choosing the proper nodes to explore next. Effective node selection is crucial for the algorithm's success in solving optimization problems. "MIP-GNN" (**MIP-GNN**) proposed by Khalil et al. (2022) uses GNN prediction to guide node selection and warm-starting, and is selected as another baseline.

### 5.2    Benchmark datasets

We evaluate the proposed framework on four ILP benchmarks with certain symmetry, which consists of (i) two problems with symmetric groups: the item placement problem (IP) and the steel mill slab problem (SMSP), (ii) the periodic event scheduling problem (PESP) with cyclic group, and (iii) a modified variant of PESP (PESPD) which has a dihedral group.

The first benchmark IP is from the NeurIPS ML4CO 2021 competition (Gasse et al., 2022). We use their source code to randomly generate instances with binary variables ranging from 208 to 1050. Each instance has a symmetric group $S_4 \sim S_{10}$. We use 500 instances for our experiments, taking 400 as the training set and the remaining 100 for testing. The SMSP benchmark is from Schaus et al. (2011), and contains 380 problem instances. We randomly select 304 of them as training data and take the others as testing data. The instances of this benchmark have 22k~24k binary variables and nearly 10k constraints, with each of them having a symmetric group $S_{111}$. The last two benchmarks are from

PESPlib Goerigk (2012), a collection of periodic timetabling problems inspired by real-world railway timetabling settings. Since PESPlib only provides a few instances, which are not sufficient to support neural network training, we randomly perturb the weights of the provided instances to generate more data (see Appendix G.3.1 for details). We respectively generate 500 instances for PESP and PESPD, taking 400 of them as training sets and 100 as testing sets. The symmetry groups of these two datasets are cyclic groups $C_5 \sim C_{15}$ and dihedral groups $D_5 \sim D_{15}$, respectively. For all training sets, 30% instances are used for validation. The average numbers of variables and constraints, as well as the symmetry groups of each benchmark problem, are summarized in Appendix F.1. Besides, more details about their ILP formulations and corresponding symmetries are supplemented in Appendix G.

These benchmarks only include problem instances. We collect the corresponding solutions using an ILP solver CPLEX (IBM, 2020). However, solving ILP instances even with moderate sizes to optimal is extremely expensive. It is more practical to use high-quality solutions as the labels. Therefore, we run single-thread CPLEX for a time limit of 3,600 seconds and record the best solutions.

## 5.3 Training settings

All models are trained with a batch size 16 for 50 epochs. The Adam optimizer with a learning rate of 0.001 is used, and other hyperparameters of the optimizer are set to their default values. The model with the smallest loss on the validation set is used for subsequent evaluations. Other training settings, such as the loss function and neural architectures, follow the configurations in Han et al. (2023). More details about the hyper-parameter tuning for the downstream tasks and software resources are shown in Section E.

## 5.4 Evaluation metrics

To compare the prediction performance of the model trained on $r$ and $r_s$, we define the *Top-m% error* for evaluation. In addition, another criterion *relative primal gap* is used to evaluate the final performance in identifying feasible solutions in different downstream tasks.

**Top-$m$% error:** We use the distance between a rounded prediction and its nearest equivalent solution as the error. Specifically, given a prediction $\hat{y}$ and its label $y$, we define the equivalent solution closest to $\hat{y}$ as $\tilde{y} = \pi'(y)$, where $\pi' = \arg\min_\pi \|\hat{y} - \pi(y)\|$. Then, the Top-$m$% error is defined as

$$\mathcal{E}(m) = \sum_{i \in M} |\text{Round}(\hat{y}_i) - \tilde{y}_i|, \tag{10}$$

where $M$ is the index set of $m$% variables with largest values of $|\text{Round}(\hat{y}_j) - \hat{y}_j|$. This error measures the minimum distance between the prediction and all solutions equivalent to the label. Compared to naive use of the distance $\sum_{i \in M} |\text{Round}(\hat{y}_i) - y_i|$, (10) can more accurately represent how close a prediction is to a feasible solution. Since for the naive distance, when $\text{Round}(\hat{y})$ equals any equivalent solution $\pi(y) \neq y$, the distance is greater than 0, while that of (10) is 0.

**Relative primal gap:** We also feed the outputs of the models trained through $r_s$ to the downstream tasks mentioned in Section 5.1 to evaluate the quality of the predictions. All the three downstream approaches incorporate ILP solvers to search for solutions. We run these ILP solvers on a single thread for a maximum of 800 seconds. Since all the problems used in the experiments are NP-hard, identifying optimality is highly time-consuming. Thus the metric used in our experiments is *relative primal gap*

$$\text{PG}(\tilde{y}) = \frac{|c^\top \tilde{y} - c^\top y^*|}{|c^\top y^*| + \epsilon}, \tag{11}$$

which measures the relative gap in the objective value of a feasible solution $\tilde{y}$ to that of the best-known solution $y^*$, and $\epsilon$ is a small positive value to avoid the numerical issue. Additionally, let $\gamma_r$ and $\gamma_{r_s}$ respectively be the primal gaps of models trained through $r$ and $r_s$, then an improvement gain of our approach is calculated as $(\gamma_r - \gamma_{r_s})/\gamma_r$.

# 6 Numerical results

In this section, we present the comparison results on empirical risk $r$ and symmetry-aware one $r_s$. In addition, primal gaps of SymILO and baselines on three downstream tasks are reported.

## 6.1 On empirical risks and Top-$m\%$ error

We denote training and test risks by $r^{tr}(\cdot) = r(\cdot; \mathcal{D}^{tr})$ and $r^{te} = r(\cdot; \mathcal{D}^{te})$, respectively, and similarly use $r_s^{tr}$ and $r_s^{te}$ for symmetry-aware risk. Let $f^{(k)}$ and $f_s^{(k)}$ be the best classic model and symmetry-aware model obtained at $k$-th epoch by training with $r^{tr}$ and $r_s^{tr}$, respectively. We plot both the training and test risks versus the number of epochs in Figure 3. As predicted in Proposition 4.1, when algorithms converge, the classic empirical risk $r^{tr}$ is always greater than symmetry-aware risk $r_s^{tr}$.

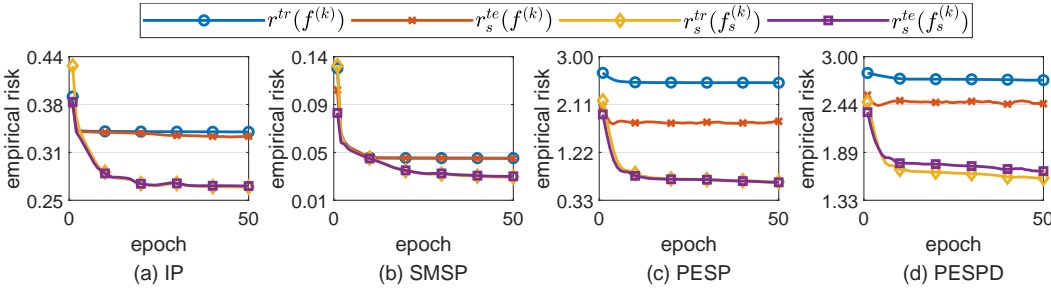

Figure 3: The training and test risks v.s. the number of epochs on four benchmark problems.

As shown in Table 1, the symmetry-aware model $f_s^{(K)}$ always predicts smaller Top-$m\%$ errors in (10) compared to the classic model $f^{(K)}$, demonstrating the usefulness of proposed empirical symmetry-aware risk in predicting solutions correctly.

Table 1: Top-$m\%$ errors ($\downarrow$) of model $f^{(K)}$ and $f_s^{(K)}$ averaged over different datasets.

| $m\%$ | IP | | SMSP | | PESP | | PESPD | |
|---|---|---|---|---|---|---|---|---|
| | $f^{(K)}$ | $f_s^{(K)}$ | $f^{(K)}$ | $f_s^{(K)}$ | $f^{(K)}$ | $f_S^{(K)}$ | $f^{(K)}$ | $f_s^{(K)}$ |
| 10% | 0.8 ±0.8 | **0.4** ±0.6 | 0.6± 0.7 | **0.0**± 0.0 | 7.5±17 | **0.1**±0.2 | 87.4±41 | **12.7**±4.8 |
| 30% | 3.9 ±1.5 | **2.9** ±1.3 | 5.3± 2.6 | **0.1**± 0.1 | 44.2±35 | **0.1**±0.5 | 275±73 | **81.3**±24 |
| 50% | 17.0 ±2.4 | **5.1** ±1.7 | 19.5± 5.5 | **0.6**± 2.5 | 52.7±35 | **0.3**±0.8 | 422±102 | **223**±77 |
| 70% | 46.5 ±2.8 | **36.3** ±4.3 | 47.5± 9.8 | **17.8**± 6.6 | 122±26 | **23**±5.5 | 638±93 | **486**±114 |
| 90% | 82.9 ±1.5 | **76.1** ±3.0 | 103± 15 | **47.0**± 9.1 | 1.6k±30 | **212**±23 | 854±69 | **848**±99 |

Moreover, the time costs of minimizing different empirical risks $r$ and $r_s$ for a mini batch are shown in Table 2. Here, $t$ denotes the average time of solving the permutation decisions per instance. The reported times for $r_s$ include the optimization time $t$. The table illustrates that the alternate training strategy does not significantly increase the training duration, and the optimization step over $\pi$ is executed efficiently.

Table 2: Time cost for minimizing different empirical risks (in seconds).

| | IP | SMSP | PESP | PESPD |
|---|---|---|---|---|
| $r$ | 5.54 | 69.43 | 14.97 | 16.17 |
| $r_s$ | 6.01 | 71.5 | 15.14 | 16.46 |
| $t$ | 0.029 | 0.129 | 0.011 | 0.018 |

## 6.2 Downstream results

The relative primal gaps of different downstream tasks at different solving time are shown in Figure 4, and the final values at 800 seconds are listed in Table 3. As Figure 4 shows, our proposed empirical risk significantly improves the performance of different downstream tasks over the primal gap in 800 seconds.

Note that the node selection task exhibits modest performance in comparison to other tasks; a possible reason is that it requires runtime interaction to call the callback functions provided by the CPLEX

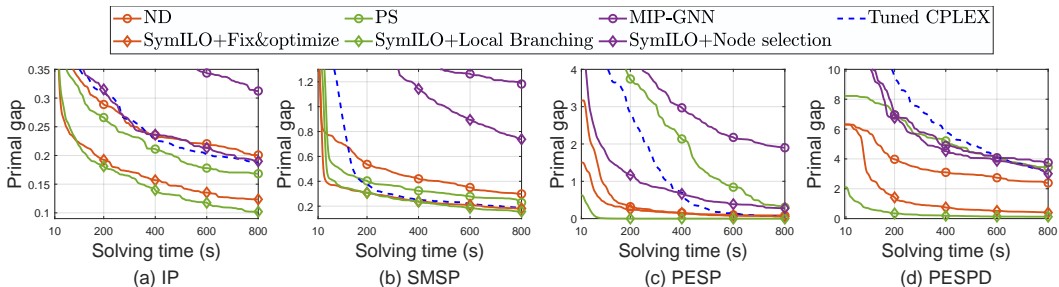

Figure 4: Relative primal gaps at different times. Three downstream tasks, i.e., fix-and-optimize, local branching, and node selection, are evaluated with a time limit of 800 seconds. The results of the same downstream task use the same color. In addition, the relative primal gap of the Tuned CPLEX running on a single thread is also reported as the blue dashed line.

Python APIs, which can slow down the whole solving process. However, such a flaw does not affect the demonstration of the effectiveness of our proposed method.

For the primal gap at 800 seconds shown in Table 3, the models trained through $r_s$ significantly improve all downstream tasks. The performance gain of the model trained through $r_s$ is calculated by computing the relative gaps between our approach's gap improvements and that of the baselines. Average gains over the three downstream tasks are 50.3%, 66.5% and 45.4%, respectively. The overall results demonstrate the effectiveness of the proposed empirical risk $r_s$. We also provide the corresponding p-values for the significance of improvements in Appendix F.2.

Table 3: Average relative primal gaps (↓) of different downstream tasks at 800 second. The values in this table are averaged over primal gaps of all test data for each benchmark problem. "Tuned CPLEX" is the result of the tuned CPLEX running on a single thread.

| Dataset | Tuned CPLEX | Fix&optimize | | | Local branching | | | Node Selection | | |
|---|---|---|---|---|---|---|---|---|---|---|
| | | ND | SymILO | gain(↑) | PS | SymILO | gain(↑) | MIP-GNN | SymILO | gain(↑) |
| IP | 0.188 | 0.201 | **0.124** | 38.4% | 0.168 | **0.102** | 39.4% | 0.312 | **0.190** | 39.2% |
| SMSP | 0.190 | 0.300 | **0.180** | 40.0% | 0.230 | **0.160** | 30.4% | 1.180 | **0.740** | 37.3% |
| PESP | 0.056 | 0.084 | **0.050** | 39.8% | 0.306 | **0.000** | 100% | 1.899 | **0.280** | 85.3% |
| PESPD | 3.194 | 2.389 | **0.404** | 83.1% | 3.442 | **0.127** | 96.3% | 3.755 | **3.006** | 20% |
| Avg. | | | | 50.3% | | | 66.5% | | | 45.4% |

## 7 Limitations and conclusions

In conclusion, we propose SymILO, a novel symmetry-aware learning framework for enhancing the prediction of solutions for integer linear programs by incorporating symmetry into the training process. Our approach shows significant performance improvements over symmetry-agnostic methods on benchmark datasets. Despite the significant advancements presented in our symmetry-aware learning framework, SymILO, several limitations must be acknowledged. Firstly, while we provide realizations for three commonly encountered symmetry groups—symmetric, cyclic, and dihedral—the framework requires specific formulations for optimizing permutations, which limits its immediate applicability to other symmetry groups not discussed in this work. Secondly, for large-scale problem instances with extensive and complex symmetry groups, the sub-problems involved in optimizing permutations can significantly slow down the training process. Enhancing the computational efficiency of our alternating optimization algorithm for these cases remains a challenge and an area for future research.

## Acknowledgments

This work was supported by the National Key R&D Program of China under grant 2022YFA1003900. Akang Wang also acknowledges support from the National Natural Science Foundation of China (Grant No. 12301416), the Shenzhen Science and Technology Program (Grant No. RCBS20221008093309021), the Guangdong Basic and Applied Basic Research Foundation (Grant No. 2024A1515010306) and the Longgang District Special Funds for Science and Technology Innovation (LGKCSDPT2023002). Ruoyu Sun also acknowledges support from the Hetao Shenzhen-Hong Kong Science and Technology Innovation Cooperation Zone Project (No. HZQSWS-KCCYB-2024016), the University Development Fund (UDF01001491) at the Chinese University of Hong Kong, Shenzhen, the Guangdong Provincial Key Laboratory of Mathematical Foundations for Artificial Intelligence (2023B1212010001), and the Guangdong Major Project of Basic and Applied Basic Research (2023B0303000001). Tsung-Hui Chang acknowledges support from the Shenzhen Science and Technology Program (Grant No. ZDSYS20230626091302006).

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

# A  Theoretical proofs

## A.1  Proposition 4.1

**Non-rigorous Proof:** Consider the case where $s_1 = s_2$, $y_1 \neq y_2$, and $\ell$ is the mean squared error, and let $\hat{y} = f_\theta(s_1) = f_\theta(s_2)$. Then,

$$r^* \geq \min_{\hat{y}} \frac{1}{2} \left( \|\hat{y} - y_1\|^2 + \|\hat{y} - y_2\|^2 \right) = \frac{1}{4}\|y_1 - y_2\|^2 > 0.$$

While for the symmetry-aware risk, since $s_1$ and $s_2$ corresponds to identical instances, there must exist permutations $\pi_1'$ and $\pi_2'$, such that $\pi_1'(y_1) = \pi_2'(y_2)$. Consequently,

$$r_s^* = \min_{\hat{y}} \|\hat{y} - \pi_1'(y_1)\|^2 + \|\hat{y} - \pi_2'(y_2)\|^2 = 0 < r^*.$$

## A.2  Proposition 4.2

**Proof**: When the loss function $\ell(\cdot, \cdot)$ is the squared error (SE) or the binary cross-entropy loss (BCE),

$$\ell_{SE}(\hat{X}, XP) = \|\hat{X} - XP\|_F^2 \tag{12}$$

$$= \text{tr}(\hat{X}^\top \hat{X} - 2\hat{X}^\top XP + P^\top X^\top XP) \tag{13}$$

$$= \text{tr}(\hat{X}^\top \hat{X} - 2\hat{X}^\top XP + X^\top X), \tag{14}$$

$$\ell_{BCE}(\hat{X}, XP) = -\sum_{j,k}([XP]_{jk} \log \hat{X}_{jk} + (1 - [XP]_{jk}) \log(1 - \hat{X}_{jk})), \tag{15}$$

equations (13) to (14) hold for permutations of a matrix's rows and columns don not change its Frobenius norm, i.e., $\text{tr}(P^\top X^\top XP) = \|XP\|_F^2 = \|X\|_F^2 = \text{tr}(X^\top X)$. (14) and (15) show that these two loss functions are linear w.r.t $P$, with which (8) becomes a linear assignment problem Kuhn (1955). It is easy to verify that the constriant matrix of a linear assignment problem is totally unimodular—-it satisfies the four conditions of Hoffman and Gale (see Page 252 in Dantzig (1956)), thus an optimal solution of the relaxed problem (9) must be integral as well, namely an optimal solution to problem (8).

# B  ILP examples with different symmetry group

***Example*** B.0.1.  Consider a bin packing problem, in which there are three items $I = \{1, 2, 3\}$ with sizes $\{a_1 = 1, a_2 = 2, a_3 = 3\}$ and three identical bins $J = \{1, 2, 3\}$ with capacity $B = 3$. Items are packed into bins, and it is required to use a minimum number of bins without exceeding the capacity. The specific formulation is as follows:

$$\min_{x_{ij}, y_j \in \{0,1\}} \quad y_1 + y_2 + y_3$$

$$a_1 x_{1j} + a_2 x_{2j} + a_3 x_{3j} \leq B y_j, \qquad \forall j \in J \tag{16a}$$

$$x_{i1} + x_{i2} + x_{i3} = 1, \qquad \forall i \in I \tag{16b}$$

where $y_j = 1$ denotes $j$-th bin is used and $x_{ij} = 1$ denotes $i$-th item is placed in $j$-th bin.

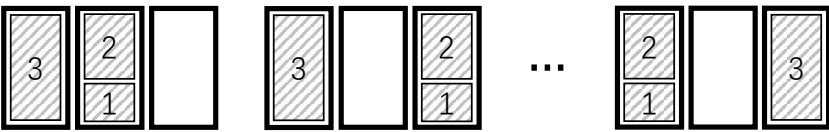

Figure 5: Equivalent solutions of Example B.0.1.

Since all bins are identical, arbitrarily swapping them does not change the feasibility and the objective value, e.g., the different assignments shown in Figure 5 are all equivalent. Specifically, assume

$$X \triangleq \begin{bmatrix} y_1 & y_2 & y_3 \\ x_{11} & x_{12} & x_{13} \\ x_{21} & x_{22} & x_{23} \\ x_{31} & x_{32} & x_{33} \end{bmatrix} = \begin{bmatrix} 1 & 1 & 0 \\ 0 & 1 & 0 \\ 0 & 1 & 0 \\ 1 & 0 & 0 \end{bmatrix}$$

is an optimal solution to the problem (16), then

$$\mathcal{X} = \left\{ \begin{bmatrix} 1 & 0 & 1 \\ 0 & 0 & 1 \\ 0 & 0 & 1 \\ 1 & 0 & 0 \end{bmatrix}, \begin{bmatrix} 1 & 1 & 0 \\ 1 & 0 & 0 \\ 1 & 0 & 0 \\ 0 & 1 & 0 \end{bmatrix}, \begin{bmatrix} 1 & 0 & 1 \\ 1 & 0 & 0 \\ 1 & 0 & 0 \\ 0 & 0 & 1 \end{bmatrix}, \begin{bmatrix} 0 & 1 & 1 \\ 0 & 0 & 1 \\ 0 & 0 & 1 \\ 0 & 1 & 0 \end{bmatrix}, \begin{bmatrix} 0 & 1 & 1 \\ 0 & 1 & 0 \\ 0 & 1 & 0 \\ 0 & 0 & 1 \end{bmatrix} \right\}$$

are all equivalent solutions to $X$. Formally, this problem has a symmetric group $S_3 = \{(1,2,3),(1,3,2),(2,1,3),(2,3,1),(3,1,2),(3,2,1)\}$ w.r.t. its bin numbers $J$.

***Example*** B.0.2. Given a circle with circumference 8, place 3 ticks at integer points around the circle such that all distances between inter-ticks along the circumference are distinct. The formulation of this problem is as follows:

$$\min_{x_1,x_2,x_3} 0$$

$$\begin{array}{llr}
\text{s.t. } y_{ij} = |x_i - x_j|, & \forall(i,j) \in S, & \text{(17a)} \\
d_{ij} = \min\{y_{ij}, 8 - y_{ij}\}, & \forall(i,j) \in S, & \text{(17b)} \\
d_{12} \neq d_{13}, d_{12} \neq d_{23}, d_{13} \neq d_{23} & \forall(i,j) \in S, & \text{(17c)}
\end{array}$$

where $S = \{(1,2),(1,3),(2,3)\}$, $x_1, x_2, x_3 \in \{1,2,3,4,5,6,7,8\}$ denote the positions of each tick, $d_{ij}$ are distances between ticks $i$ and $j$ with auxiliary variables $y_{ij}$. The constraints in this formulation are nonlinear, we linearize them by big-M methods. Equalities (17a) can be linearized by introducing auxiliary variables $a_{ij} \in \{0,1\}, \forall(i,j) \in S$ as

$$\begin{array}{llr}
y_{ij} \geq x_i - x_j, & \forall(i,j) \in S, & \text{(18a)} \\
y_{ij} \geq x_j - x_i, & \forall(i,j) \in S, & \text{(18b)} \\
y_{ij} \leq x_i - x_j + 8 \cdot a_{ij}, & \forall(i,j) \in S, & \text{(18c)} \\
y_{ij} \leq x_j - x_i + 8 \cdot (1 - a_{ij}), & & \text{(18d)}
\end{array}$$

when $x_i \geq x_j$, $a_{ij} = 0$, otherwise $a_{ij} = 1$. Similarly, equalities (17b) are equivalent to

$$\begin{array}{llr}
d_{ij} \leq y_{ij}, & \forall(i,j) \in S, & \text{(19a)} \\
d_{ij} \leq 8 - y_{ij}, & \forall(i,j) \in S, & \text{(19b)} \\
d_{ij} \geq y_{ij} - 8 \cdot m_{ij}, & \forall(i,j) \in S, & \text{(19c)} \\
d_{ij} \geq 8 - y_{ij} - 8 \cdot (1 - m_{ij}), & \forall(i,j) \in S, & \text{(19d)}
\end{array}$$

where $m_{ij} \in \{0,1\}, \forall(i,j) \in S$ are auxiliary variables, with $m_{ij} = 0$ when $y_{ij} \leq 8 - y_{ij}$.

The not-equal constraints (17c) can be linearized by

$$\begin{array}{llr}
d_{ij} \geq d_{k\ell} + 1 - 8 \cdot t_{ijk\ell}, & \forall(i,j,k,\ell) \in K, & \text{(20a)} \\
d_{k\ell} \geq d_{ij} + 1 - 8 \cdot (1 - t_{ijk\ell}), & \forall(i,j,k,\ell) \in K, & \text{(20b)}
\end{array}$$

where $K = \{(1,2,1,3),(1,2,2,3),(1,3,2,3)\}$. By introducing auxiliary variables $t_{ijk\ell} \in \{0,1\}, \forall(i,j,k,\ell) \in K$, we have $d_{ij} \geq d_{k\ell} + 1$ if $t_{ijk\ell} = 1$, otherwise $d_{ij} \leq d_{k\ell} - 1$, i.e., $d_{ij} \neq d_{k\ell}$.

Assume $\{x_1 = \bar{x}_1, x_2 = \bar{x}_2, x_3 = \bar{x}_3\}$ is a feasible solution of this problem and let $[\cdot]_T$ denote the $mod - T$ operation, then it's easy to verify that $\{x_i = [\bar{x}_i + b]_8\}_{i=1}^3$ (rotation) and its reverse $\{x_i = [(8 - \bar{x}_i) + b]_8\}_{i=1}^3, \forall b \in \mathbb{Z}$ (reflection of the rotation) are both equivalent feasible solutions. It is more intuitive to see the illustration in Figure 6, rotation and reflection acting on the ticks do not change their corresponding distances.

When representing $x_1, x_2, x_3$ by binary variables $z_{ip} \in \{0,1\}, \forall i \in 1,2,3, \forall p \in \{1,\ldots,8\}$:

$$x_i = \sum_{p}^{8} p \cdot z_{ip}, \qquad\qquad \forall i \in \{1,2,3\}, \qquad\qquad \text{(21a)}$$

$$\sum_{p}^{8} z_{ip} = 1, \qquad\qquad \forall i \in \{1,2,3\} \qquad\qquad \text{(21b)}$$

the modulo symmetry leads to a dihedral group $D_8$ along the $p$ dimension of $z_{ip}$. Specifically, let $Z$ be a feasible solution with its $(i, p)$-th entry as the value of $z_{ip}$, then any permutation $\pi \in D_8$ acting on the columns of $Z$ yields another equivalent solution $\left[ Z_{:\pi(1)}, \ldots, Z_{:\pi(8)} \right]$.

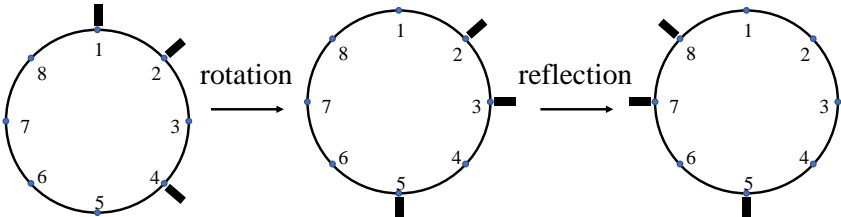

Figure 6: Equivalent solutions of Example B.0.2.

## C  Bipartite graph representation for MILP

Gasse et al. (2019) proposed to represent a MILP (also works for ILP) by a bipartite graph $\mathcal{G} = (V, C, E)$ with two disjoint sets of nodes, $V = \{v_1, v_2, \ldots, v_n\}$ and $C = \{c_1, c_2, \ldots, c_m\}$, denoting the decision variables and constraints in Problem (1), respectively. And $E = \{A_{jk} | A_{jk} \neq 0, c_j \in C, v_k \in V\}$ is the set of weighted edges connecting variable nodes and constraint nodes, where $A$ is the coefficient matrix in Problem (1). Each node has a feature vector $v_k$ or $c_j$ describing the information of the variables or constraints. For example, in our experiments, these features include variable types (continuous or binary), variable positions, lower and upper bounds, right-hand side coefficients, constraint types $(=, \leq, \geq)$, etc.

## D  Graph convolutional neural network

In the graph convolutional neural network (GCNN)-based approach proposed by Gasse et al. (2019), a bipartite graph $\mathcal{G} = (V, C, E)$ (with node features $c_j^{(0)} = c_j, v_k^{(0)} = v_k$, and edge features $A_{jk}$) is taken as the input. Stacked layers are applied to aggregate information from neighbors and update node embeddings. Each layer has two consecutive *half convolutions* computed as

$$c_j^{(l)} = f_c^{(l)} \left( c_j^{(l-1)}, \sum_{j:(j,k)\in E} g_c^{(l)} \left( c_j^{(l-1)}, v_k^{(l-1)}, A_{jk} \right) \right), \tag{22}$$

$$v_k^{(l)} = f_v^{(l)} \left( v_k^{(l-1)}, \sum_{k:(j,k)\in E} g_v^{(l)} \left( c_j^{(l)}, v_k^{(l-1)}, A_{jk} \right) \right), \tag{23}$$

where $l \in \{0, \cdots, L\}$ denotes the layer index, $f_c^{(l)}$ and $g_c^{(l)}$ are non-linear transformations gathering information from variable nodes and update on constraint nodes, $f_v^{(l)}$ and $g_v^{(l)}$ are on the contrary. All of these four transformations are two-layer perceptrons with ReLU activations. Lastly, another two-layer perceptron $f_{out}$ with sigmoid activation is used to convert the final embeddings to the predictions of integer variables by $\hat{x}_k = \text{Sigmoid}(f_{out}(v_k^{(L)}))$. We denote $f_\theta$ as the GNN parameterized by $\theta$, and the output vector for the discrete part as $\hat{x} = f_\theta(\mathcal{G})$ in the remaining sections.

## E  Detailed experimental settings

### E.1  Hyper-parameter tuning

The experiments involved three downstream tasks: fix&optimize, local branching, and node selection. For the first two, we utilized grid search for hyperparameter tuning. In fix&optimize, we adjusted $\alpha$, the fraction of variables to fix, exploring values from 0.1 to 0.9. For local branching, we varied $\beta$, the percentage of variables in the local branching constraint, within the same range. For node selection, we adhered to default settings as stated in Khalil et al. (2022). In Table E.1, we report the optimal

hyperparameters for each task and dataset, ensuring clarity and aiding in the reproducibility of our work.

Table 4: Hyper-parameters for different down-stream tasks

| Dataset | $r$ | | $r_s$ | |
|---|---|---|---|---|
| | $\alpha$ | $\beta$ | $\alpha$ | $\beta$ |
| IP | 0.1 | 0.2 | 0.1 | 0.1 |
| SMSP | 0.5 | 0.5 | 0.5 | 0.6 |
| PESP | 0.1 | 0.1 | 0.3 | 0.4 |
| PESPD | 0.1 | 0.1 | 0.3 | 0.2 |

## E.2 Computational resources and software

All evaluations are performed under the same configuration. The evaluation machine has one AMD EPYC 7H12 64-Core Processor @ 2.60GHz, 256GB RAM, and one NVIDIA GeForce RTX 3080. CPLEX 22.2.0 and PyTorch 2.0.1 (Paszke et al., 2019) are utilized in our experiments. The time limit for running each experiment is set to 800 seconds since a tail-off of solution qualities was often observed after that.

# F Other supplements

## F.1 Dataset details

Table 5: More information about benchmark problems include average number of variables ("bin." for bianry while "int." for integer) and constraints, as well as symmetry groups.

| problem | # of Var. | # of Cons. | symmetry |
|---|---|---|---|
| IP | $208 \sim 1050$ bin. | $46 \sim 196$ | $S_4 \sim S_{10}$ |
| SMSP | $22k \sim 24k$ bin. | $20k \sim 22k$ | $S_{111}$ |
| PESP | $5k \sim 15k$ int. | $7k \sim 21k$ | $C_5 \sim C_{15}$ |
| PESPD | $5k \sim 15k$ int. | $10k \sim 30k$ | $D_5 \sim D_{15}$ |

## F.2 p-values for the significance of improvement

We use the paired-sample T-test in MATLAB and report p-values in the following table. A p-value less than 0.05 means the mean difference (improvement) is significant. It shows that our proposed framework is effective in finding a better solution.

Table 6: p-values of paired-sample T-tests for the difference of means of the relative primal gaps

| p-value | Fix and Optimize | Local Branching | Node Selection |
|---|---|---|---|
| IP | 0.0012352 | 0.0007626 | 0.0044869 |
| SMSP | 0.0000441 | 0.0007752 | 0.0000026 |
| PESP | 0.0589491 | 0.0585871 | 0.0012326 |
| PESPD | 0.0000002 | 0.0000054 | 0.0088221 |

# G Problem formulation and their corresponding symmetry group

## G.1 IP

There are $I$ items, $J$ bins, and $K$ resource types. Each item $i$ has a fixed resource requirement $a_{ik}$ for each resource type $k$. Each bin $j$ has a fixed capacity $b_k$ for each resource type $k$. The goal is to

place all items in bins, while minimizing the imbalance of the resources used across all bins. This problem has a formulation as follows:

$$\min_{x,y,z} \quad \sum_{j \in J} \sum_{k \in K} \alpha_k y_{jk} + \sum_{k \in K} \beta_k z_k$$

$$\text{s.t.} \quad \sum_{j \in J} x_{ij} = 1 \qquad\qquad \forall i \in I \qquad (24\text{a})$$

$$\sum_{i \in I} a_{ik} x_{ij} \leq b_k \qquad\qquad \forall j \in J, \forall k \in K \qquad (24\text{b})$$

$$\sum_{i \in I} d_{ik} x_{ij} + y_{jk} \geq 1 \qquad\qquad \forall j \in J, \forall k \in K \qquad (24\text{c})$$

$$y_{jk} \leq z_k \qquad\qquad \forall j \in J, \forall k \in K \qquad (24\text{d})$$

$$x_{ij} \in \{0,1\} \qquad\qquad \forall i \in I, \forall j \in J \qquad (24\text{e})$$

$$y_{jk} \geq 0 \qquad\qquad \forall j \in J, \forall k \in K \qquad (24\text{f})$$

where $x_{ij} = 1$ denotes assigning item $i$ to bin $j$, $d_{ik}$ is normalized resource requirement for each item, $y_{jk}$ and $z_k$ are implicit decision variables to track the imbalance of the resources. Since each bin in this problem is identical (i.e., with the same capacity), reordering bins would not change a feasible solution's feasibility and objective value. This problem naturally has a symmetric group $S_{|J|}$ w.r.t. the ordering of bins $J$. Specifically, let $X \in \{0,1\}^{|I| \times |J|}$ be a feasible solution of an IP instance with its $(i,j)$-th entry as the value of variable $x_{ij}$. Then arbitrary permutation $\pi \in S_{|J|}$ acting on its columns $\{X_{:j}, \forall j \in J\}$ yields an equivalent solution $\left[ X_{:\pi(1)}, X_{:\pi(2)}, \ldots, X_{:\pi(|J|)} \right]$.

## G.2 SMSP

Given order set $O$, and slab set $S$. Color set $C$, and slab weights $Q = \{u_0 = 0, u_1, u_2, ..., u_k\}$. $u_0$ denotes unused slab. The ILP formulation of SMSP used in our experiments is from (Gargani & Refalo, 2007) as

$$\min_{x,y,z} \quad \sum_{s \in S, q \in Q} q \times y_{qs}$$

$$\text{s.t.} \sum_{o \in O} x_{os} = 1 \qquad\qquad \forall s \in S, \qquad (25\text{a})$$

$$\sum_{q \in Q} y_{qs} = 1 \qquad\qquad \forall s \in S, \qquad (25\text{b})$$

$$\sum_{o \in O} w_o x_{os} \leq \sum_{q \in Q} q \times y_{qs} \qquad\qquad \forall s \in S, \qquad (25\text{c})$$

$$x_{os} \leq z_{c_o s} \qquad\qquad \forall o \in O, s \in S, \qquad (25\text{d})$$

$$\sum_{c \in C} z_{cs} \leq 2 \qquad\qquad \forall s \in S, \qquad (25\text{e})$$

$$x_{os}, y_{qs}, z_{cs} \in \{0,1\} \qquad\qquad \forall o \in O, s \in S, c \in C, q \in Q. \qquad (25\text{f})$$

## G.3 PESP

Periodic event scheduling problem involves determining optimal schedules for a set of events that occur repeatedly over a fixed period, such as bus or train departures. Consider a set of events $\mathcal{E}$. For each event $i \in \mathcal{E}$ we would like to schedule a time $t_i \in \{1, \ldots, T-1\}$, where $T$ is the periodic length. Besides, a set of activities $\mathcal{A} \subseteq \mathcal{E} \times \mathcal{E}$ connect events with each other. Each activity $a \in \mathcal{A}$ has a lower bound $\ell_a \in \mathbb{N}$, an upper bound $u_a \in \mathbb{N}$, and a weight $w_a$. The goal is to minimize the weighted sum of the slack $y_a$ of all activities, while ensuring all activity slacks are within $[0, u_a - \ell_a]$.

It can be formulated as:

$$\min_t \sum_{a \in \mathcal{A}} w_a(y_a + \ell_a)$$

$$s.t.\ y_a = [t_j - t_i]_T \qquad\qquad \forall a = (i,j) \in \mathcal{A}, \qquad (26a)$$
$$0 \le y_a \le u_a - \ell_a \qquad\qquad \forall a \in \mathcal{A}, \qquad (26b)$$
$$t_i \in \{0, \ldots, T-1\}, \qquad\qquad \forall i \in \mathcal{E} \qquad (26c)$$

where $[\cdot]_T$ denotes the modulo operation, which enforces the periodic nature. It is modeled as $[t_j - t_i]_T \triangleq t_j - t_i + z_a T$ by introducing additional implicit variables $\{z_a \in \mathbb{N}\}$. Due to the existence of modulo operation, we can regard $\{t_i, \forall i \in \mathcal{E}\}$ as frames in a clock with intervals $[0, \ldots, T-1]$. If all $t_i$ rotate the same angles simultaneously, then the activity slacks $y_a$ remain unchanged. In our experimentation, we substitute $t_i$ by $t_i = \sum_k (k-1) \cdot x_{ik}$ and $\sum_k x_{ik} = 1$, where $x_{ik} \in \{0,1\}, \forall i \in \mathcal{E}, k \in \{1, T\}$. Let $X \in \{0,1\}^{|\mathcal{E}| \times T}$ denotes a feasible solution with its $(i, k)$-th entry as the value of variable $x_{ik}$, then this problem has a cyclic group $C_T$ w.r.t. the column indices of $X$, i.e., any permutation $\pi \in C_T$ acting on the columns of $X$ yields an equivalent feasible solution $[X_{:\pi(1)}, \ldots, X_{:\pi(T)}]$.

### G.3.1 Data generation by perturbation

As mentioned above, a PESP instance has a set of events $\mathcal{E}$ and a set of activities $\mathcal{A} \subseteq \mathcal{E} \times \mathcal{E}$ connecting events with each other. Each activity has a weight $w_a$. The goal is to assign an appropriate time $t_i$ to each event $i \in \mathcal{E}$ to meet some constraints while minimizing the total time slack weighted by $\{w_a, a \in \mathcal{A}\}$. These weights heavily impact the time assignment. We perturb these weights to generate new instances by introducing Gaussian noises, i.e., $w'_a = w_a + n_a$, where $n_a \sim \mathcal{N}(\mu = w_a, \sigma = 0.1 * w_a)$.

