# OpenReview forum: "SymILO: A Symmetry-Aware Learning Framework for Integer Linear Optimization"
_NeurIPS.cc/2024/Conference — NeurIPS 2024 poster_

### Official Review · Reviewer_nCep · 2024-06-27

**Soundness:** 3
**Presentation:** 3
**Contribution:** 2
**Rating:** 6
**Confidence:** 4

**Summary:**

This paper proposes the SymILO framework for solving combinatorial optimization problems (using machine learning). Building on traditional supervised learning, this method leverages the inherent symmetry structure of the problem to guide the model in learning the true patterns. The author also show its performance on several datasets.

**Strengths:**

1. The idea of considering symmetry is interesting and crucial, especially when using the optimal solutions as labels. Since the solution space could be symmetry-invariant, using one particular solution as a label may cause the model to fail in learning the intrinsic pattern.
2. The presentation is neat.

**Weaknesses:**

See questions.

**Questions:**

1. Although the idea is good, the permutation group, which plays a crucial role, is only vaguely explained in the paper. In most typical combinatorial optimization problems, the permutation group is subtle and may even vary with the instance. In the main body of the paper (Section 4), the authors' direct use of $ G_i $ is confusing.
2. Following up on the previous point, the authors should explain how to identify the appropriate permutation group for a new problem and clarify whether selecting a specific permutation group significantly impacts the computational complexity during the optimization process.
3. Where in the paper are the details on "passing the outputs to an off-the-shelf solver" provided?
4. I expect the authors to explicitly formulate the problem as a MIP problem in the examples provided in Section B and specify the permutation group.

**Limitations:**

See questions

---

> ### Author Rebuttal · Authors · 2024-08-07
>
> Thank you for the valuable comments. We apologize for any misunderstanding caused by our presentation and will address your questions below. **Please note the top-most "author rebuttal", a brief summary, and part of the response is put there**.
>
> ___Question 1: ... the permutation group ,..., is only vaguely explained in the paper. In most typical combinatorial optimization problems, the permutation group is subtle and may even vary with the instance. In the main body of the paper (Section 4), the authors' direct use of $G_i$ is confusing.___
>
>
> _**Question 1.1:** ... the permutation group ,..., is only vaguely explained..._
>
> Some notions related to the symmetry group are additionally clarified in **part I of the global rebuttal**. Besides, concrete symmetry groups and the optimization over them for benchmark problems are specified in **part III of the global rebuttal**. Please refer to them accordingly.
>
>
> _**Question 1.2:** In most ... problems, the permutation group is subtle and may even vary with the instance. In the main body of the paper (Section 4), the authors' direct use of $G_i$ is confusing_
>
> We agree that "the permutation group is subtle and may even vary with the instance". That's also why we focus on ILP problems with specific symmetry types, instead of general ILPs.
> Instances in the same dataset share the same type of symmetry group, while the group size may vary with their instance sizes. For example, in the IP dataset, instances $\\{s_1,\dots,s_N\\}$ have bin numbers $\\{b_1,\dots,b_N\\}$, then their corresponding symmetry groups $G_i=S_{b_i},\forall i=\\{1,\dots,N\\}$.
>
>
> ___Question 2: ..., the authors should explain how to identify the appropriate permutation group for a new problem and clarify whether selecting a specific permutation group significantly impacts the computational complexity during the optimization process.___
>
>
> _**Question 2.1**: on the identification of the symmetry group_
>
> According to the problems considered, the situations for symmetry identification are different:
> - Case I: For a general ILP, its symmetry group can be efficiently detected by well-developed tools, such as Nauty[1], Bliss[2], etc.
> - Case II: For specific types of problems, such as bin packing, periodic scheduling, and job scheduling, the symmetry groups are already well-known by area experts [3].
>
> The problems considered in our experiments belong to Case II, thus the corresponding symmetry group is known. Although only three types of symmetry in Case II are considered, just as Reviewer HH73 commented, "this does include a number of important problems".
> Besides, **we remark that our framework is also applicable to Case I if the corresponding subproblems have appropriate customization and be solved**.
>
>
> _**Question 2.2**: whether selecting a specific permutation group significantly impacts the computational complexity during the optimization process._
>
>
> We do not fully understand the meaning of "selecting a specific perumtation group". In our work, the symmetry group of an ILP is its intrinsic property and should be known before trainning, hence it's not selectable. Could the reviewer clarify this a bit more?
>
> The computational complexity of the sub-problem over the symmetry group depends on its symmetry types. For the three symmetry types considered in our paper, the corresponding complexities are $O(q)$ for both a cyclic group $C_q$ and a dihedral group $D_q$, and $O(q^2log^q)$ for a symmetric group $S_q$, respectively. Here, $q$ is a number related to the sizes of the specific symmetry group.
> For other symmetry groups that are not considered in our paper, it depends on how the sub-problem over symmetry groups is customized.
>
>
>
> ___Question 3: Where in the paper are the details on "passing the outputs to an off-the-shelf solver" provided?___
>
> In the main paper, there is no "passing the outputs to an off-the-shelf solver". Does the reviewer mean "It can be done quite efficiently with the aid of off-the-shelf LP solvers, such as Gurobi Optimization, LLC (2023), CPLEX IBM (2020), etc." at line 175?
> Since the optimization over symmetric groups is equivalent to solving a Linear Program (LP) (see Proposition 4.2), we directly leave the job to an LP solver such as CPLEX and Gurobi. Given an LP, it can efficiently identify its optimal solution.
>
>
> ___Question 4: I expect the authors to explicitly formulate the problem as a MIP problem in the examples provided in Section B and specify the permutation group.___
>
> The formulation and symmetry group of Example B.0.1 has been specified in **part III of the global author rebuttal**. Due to the limited space, we put the specification of Example B.0.2. to another comment box, please refer to it.
>
>
> ---
>
> [1] McKay B D. Nauty user’s guide (version 2.4)[J]. Computer Science Dept., Australian National University, 2007: 225-239
>
> [2] Junttila T, Kaski P. Conflict propagation and component recursion for canonical labeling[C]//International Conference on Theory and Practice of Algorithms in (Computer) Systems.
>
> [3] Margot F. Symmetry in integer linear programming[J]. 50 Years of Integer Programming 1958-2008: From the Early Years to the State-of-the-Art, 2009: 647-686.

---

> > ### Author Response · Authors · 2024-08-08
> > **Question 4 (cont.)**
> >
> > Here we supplement the specification of Example B.0.2 in the Appendix.
> >
> > **1. Description:** Given a circle with circumference $8$, place $3$ ticks at integer points around the circle such that all distances between inter-ticks along the circumference are distinct.
> >
> > **2. Formulation:**
> > $$\\begin{align}
> >     \\min_{x_1,x_2,x_3} &~~~~~0 \\notag \\\\
> >     s.t.
> >     ~& y_{ij} = |x_i - x_j|, &&\\forall (i,j) \\in S,&&&(1)\\\\
> >     ~& d_{ij} = \\min \\{y_{ij},8-y_{ij}\\}, &&\\forall (i,j) \\in S,&&&(2)\\\\
> >     ~& d_{12} \\neq d_{13}, d_{12} \\neq d_{23}, d_{13} \\neq d_{23},&& &&&(3)
> > \\end{align}$$
> > where $S = \\{(1,2),(1,3),(2,3)\\}$, $x_1,x_2,x_3 \\in \\{1,2,3,4,5,6,7,8\\}$ denote the positions of each tick,  $d_{ij}$ are distances between ticks $i$ and $j$ with auxiliary variables $y_{ij}$.
> >
> > **3. Linearization of nonlinear constraints:**
> > The constraints in the above formulation are nonlinear, we linearize them by big-M methods.
> > - **Constraints (1):**
> > Equalities $(1)$ can be linearized by introducing auxiliary variables $a_{ij} \\in \\{0,1\\} , \\forall (i,j)\\in S$ as
> > $$
> >     \\begin{align}
> >     &y_{ij} \\geq x_i - x_j, &\\forall (i,j) \\in S,&&&(4)\\\\
> >     &y_{ij} \\geq x_j - x_i, &\\forall (i,j) \\in S,&&&(5)\\\\
> >     &y_{ij} \\leq x_i - x_j + 8\\cdot a_{ij}, &\\forall (i,j) \\in S,&&&(6)\\\\
> >     &y_{ij} \\leq x_j - x_i + 8\\cdot(1-a_{ij}),&\\forall (i,j) \\in S,&&&(7)
> >     \\end{align}
> > $$
> > where $a_{ij} = 0$ enforces $x_i-x_j\\geq0$, otherwise $x_i-x_j\\leq0$.
> >
> > - **Constraints (2):**
> > Similarly, equalities $(2)$ are equivalent to
> > $$
> > \\begin{align}
> > &d_{ij} \\leq y_{ij}, &\\forall (i,j) \\in S,\\\\
> > &d_{ij} \\leq 8-y_{ij}, &\\forall (i,j) \\in S,\\\\
> > &d_{ij} \\geq y_{ij} - 8\\cdot m_{ij}, &\\forall (i,j) \\in S,\\\\
> > &d_{ij} \\geq 8-y_{ij} - 8\\cdot(1-m_{ij}), &\\forall (i,j) \\in S,
> > \\end{align}
> > $$
> > where $m_{ij}\\in\\{0,1\\}, \\forall (i,j) \\in S$ are auxiliary variables, with $m_{ij}=0$ when $y_{ij}\\leq 8-y_{ij}$.
> >
> > - **Constraints (3):**
> > The "not equal to" constraints $(3)$ can be linearized by
> > $$
> > \\begin{align}
> >     &d_{ij} \\geq d_{k\\ell}+1 - 8\\cdot t_{ijk\\ell}, &\\forall (i,j,k,\\ell) \\in K,\\\\
> >     &d_{k\\ell} \\geq d_{ij}+1 - 8\\cdot(1- t_{ijk\\ell}), &\\forall (i,j,k,\\ell) \\in K,
> > \\end{align}
> > $$
> > where $K = \\{(1,2,1,3),(1,2,2,3),(1,3,2,3)\\}$. By introducing auxiliary variables $\\{t_{ijk\\ell}\\in \\{0,1\\}, \\forall (i,j,k,\\ell) \\in K\\}$, we have $d_{ij} \\geq d_{k\\ell} + 1$ if $t_{ijk\\ell}=1$, otherwise $d_{ij} \\leq d_{k\\ell}-1$, i.e., $d_{ij}\\neq d_{k\\ell}$.
> >
> > **4. Symmetry group:**
> >
> > Assume $\\{x\_1=\\bar{x}\_1,x\_2=\\bar{x}\_2,x\_3=\\bar{x}\_3\\}$ is a feasible solution of this problem and let $[\\cdot]_T$ denote the $mod-T$ operation, then it's easy to verify that $\\{x\_i=[\\bar{x}\_i+b]_8\\}\_{i=1}^3$ (rotation) and its reverse $\\{x_i=[(8-\\bar{x}_i)+b]_8\\}\_{i=1}^3, \\forall~ b\\in \\mathbb{Z}$ (reflection) are both equivalent feasible solutions. That is, rotation and reflection acting on the ticks do not change their corresponding distances (please refer to Figure 6 of the Appendix for visual illustration).
> >
> > When representing $x_1,x_2,x_3$ by binary variables $z_{ip}\\in \\{0,1\\}, \\forall i\\in{1,2,3},\\forall p\\in\\{1,\\dots,8\\}$:
> >
> > \\begin{align}
> >     &x_i = \\sum_{p=1}^8 p \\cdot z_{ip}, & \\forall i \\in \\{1,2,3\\},\\\\
> >     &\\sum_{p=1}^8 z_{ip} = 1, &\\forall i \\in \\{1,2,3\\},
> > \\end{align}
> >
> > the modulo symmetry leads to a dihedral group $D_8$ along the $p$ dimension of $z_{ip}$. Specifically, let $Z$ be a feasible solution with its $(i,p)$-th entry as the value of $z_{ip}$, then any permutation $\\pi \\in D_8$ acting on the columns of $Z$ yields another equivalent solution $\\left[ Z_{:\\pi(1)},\\dots,Z_{:\\pi(8)} \\right]$.

---

> > > ### Comment · Reviewer_nCep · 2024-08-08
> > > **Response**
> > >
> > > > Question 2.2: whether selecting a specific permutation group significantly impacts the computational complexity during the optimization process.
> > >
> > > I want to ask when the permutation group is large (e.g. $k!$ ), how to select the representative permutations?
> > >
> > > The reviewer answered my question about the choice of permutation group. I have raised the score. I would like the author to put these in the revised paper. It is a nice idea as long as it is decently implemented.

---

> > > > ### Author Response · Authors · 2024-08-08
> > > > **Reply to Question 2.2 (cont.)**
> > > >
> > > > Thank you for your feedback and support. We apologize for misunderstanding your question and provide the following clarification:
> > > >
> > > > As stated in the previous response to Question 2, **the complexity of selecting an optimal permutation from a symmetry group depends on how the sub-problem is customized**.
> > > >
> > > > For example, in our paper, the symmetric group (denote it as $S_k$) is a large group with $k!$ elements. Despite its large search space, we designed a sub-problem (**which is a linear program**) to efficiently select the optimal permutation (please refer to Proposition 4.2 and its proof). **The complexity of solving such a sub-problem is $O(q^2\text{log}q)$**, and can be easily done in our experiments (please refer to Table 2 in the manuscript).
> > > >
> > > > Thank you again for the valuable comments and suggestions, we will add the corresponding clarifications to the revised manuscript.

---

### Official Review · Reviewer_cWBm · 2024-07-02

**Soundness:** 3
**Presentation:** 3
**Contribution:** 3
**Rating:** 6
**Confidence:** 5

**Summary:**

The article examines the problem of predicting solutions to MILPs with a large number of symmetric solutions. An integer linear program (ILP) is symmetric if its variables can be permuted without changing the structure of the problem. The authors propose a new formulation of supervised learning for problems with symmetry. It includes choosing the optimal permutation of the target solution in the training set. The authors provide a new algorithm that updates learning parameters and permutations alternately. The proposed paradigm can be applied to complement existing methods in which the output of the model is the prediction of the solution. Experiments are conducted together with SL methods for node selection, local branching, and Neural Diving.

**Strengths:**

1.	The motivational part is well explained, and examples of symmetry are provided to understand this problem. The theoretical preliminaries provide the reader with the necessary information to understand the symmetry group, permutation classes, etc.

2.	The framework is theoretical and general, since it reformulates the supervised learning objective. Hence, it is applicable to any SL-based task, where the target is a solution for combinatorial optimization problem. It also contains proofs that symmetry-aware risk is theoretically preferable, if there exist permutations of instance’s optimal solutions.

3.	The experimental design is solid, since the proposed method is combined with three popular deep learning-based frameworks (Neural diving, Node selection with GNNs, Predict-and-search) and for each of them shows a significant increase in quality when solving problems with a large number of symmetric solutions.

**Weaknesses:**

1.	There exist classical approaches (Dantzig-Wolfe decomposition, orbital branching, and etc) that are applied to tackle the symmetry in MILPs. So, the discussion if some of them can be theoretically utilized together with the proposed framework, or some comparison (for example ND + symILO vs ND + some classical method for symmetric problems) would benefit the paper.

2.	I would also suggest testing performance on other problems (probably with less symmetry) and comparing the computational time. This would trigger a discussion about whether the proposed structure should be applied if the degree of symmetry is unknown.

3.	In limitations, the authors say that the sub-problems involved in optimizing permutations can significantly slow down the training process for large-scale problems. An analysis of how significant it will be is needed, as well as an analysis of the convergence and scalability of the alternating minimization algorithm (4.2).

**Questions:**

1.	In the vast majority of real cases, a symmetry group G is not known, we know only one of its subgroups. It would be interesting to compare the effect of adding a search for an optimal permutation over a symmetry group, with the same search over some subgroup.

2.	How to check the symmetry of the new problems for real-world scenarios, if one wants to apply the proposed symmetry-aware framework? Does it work for non-symmetry problems? Prop 4.1 holds only for the known symmetry group.

3.	Should one apply the algorithm for symmetric problems where the number and the size of similar solutions is unknown?

4.	how to compute \pi^' in line 267? Is argmin_\pi a trivial problem?

5.	Refer to weaknesses

**Limitations:**

The authors point to the potential slowdown on a large instance as a limitation. However, authors do not provide deeper analysis of it.

---

> ### Author Rebuttal · Authors · 2024-08-07
>
> Thank you for the detailed review and valuable comments. We address each weakness and question below.
>
> __W1: Applying classical approaches.__
>
> Thank you for raising the insightful comments. Below is our clarification.
>
> Our framework includes two parts.
> - a learning part: a GNN model that predicts an initial solution.
> - a post-processing part: different downstream approaches can be equipped in this part to identify high-quality solutions based on the initial solution.
>
> We argue that 1) classical symmetry-handling methods can not be directly applied in the *learning part*.
> - Symmetry-handling techniques such as orbital branching are often applied during the branch-and-bound process, while Dantzig-Wolfe decomposition reformulates problems for tighter relaxations. Both approaches are beyond the scope of the learning framework.
> - For future research, it is interesting to learn ideas from classical symmetry-handling methods and design appropriate learning algorithms.
>
> 2) Most downstream approaches already utilized classic symmetry-handling methods.
> - All downstream approaches considered in our work have the aid of ILP solvers. For example, "fix&optimize" fixes variables based on GNN's predictions, and solves sub-ILPs via an ILP solver.
> - ILP solvers, such as CPLEX, have integrated symmetry-handling methods (e.g., orbital branching/fixing). Besides, parameters are tuned to properly handle symmetry. Please refer to our response to Question 6 of Reviewer HH73.
>
> __W2: Testing on other problems.__
>
> Thank you for the valuable comment.
>
> We extend our experiments with two additional datasets with less symmetry. One is "Workload Appointment" (WA) from [1], and the other is from "assign" problems in MIPLIB 2017 (AP). We report the averaged **primal gaps** for each algorithm, alongside with LP solving time and the degree of symmetry.
>
> |Dataset||Tuned CPLEX||PS||SymILO|LP time|| $\text{log}\|G\|$|gain($\uparrow$)|
> |-|-|-|-|-|-|-|-|-|-|-|
> |WA||0.08||0.02||**0.02**|0.000||0|0.00%|
> |AP||0.72||0.41||**0.34**|0.011||4.54|17.0%|
> |IP||1.14||0.97||**0.58**|0.029||15.1|39.4%|
>
> From the above table, we have some observations:
> - Problems with more symmetry can benefit more when using SymILO.
> - Our method can also be applied to no-symmetry problems (WA), and has a comparable performance to the baseline PS.
> - Although more symmetry requires longer LP solving time, it is not a bottleneck on the considered datasets.
>
> If the symmetry is unknown, we need some well-developed tools such as Nauty [2] and Bliss [3] to detect it. Otherwise, our method would turn into a "symmetry-agnostic" one, and would not bring improvements compared to classic ones.
>
> __W3: Scalability analysis.__
>
> 1. complexity of the sub-problems over the symmetry group
>
>     1.1. The complexity of different symmetry groups.
>     - cyclic group $C_q$ and dihedral group $D_q$:  $O(q)$.
>     - symmetric group $S_q$: $O(q^2\text{log}q)$ (a linear assignment problem).
>
>     1.2. Our claim in the limitation part comes from $O(q^2\text{log}q)$ for the symmetric group. The training process can be slower when solving larger sub-problems.
>
> 2. analysis of the convergence and scalability of the alternating minimization algorithm
>
>     2.1. convergence
>
>     - Empirically, the alternating algorithm could converge in our experiments (as shown in Figure 3 in the manuscript).
>     - Theoretically, existing convergence conclusions can not be directly applied due to the discrete nature of sub-problems over symmetry groups.
>
>     2.2. scalability
>
>     - Compatibility to variable sizes: benefiting from the message-passing mechanism of GNNs, our alternating algorithm can be applied to ILPs with different sizes.
>     - Computational complexity: as mentioned before, the complexity of our algorithm depends on the sub-problems over the symmetry group. In our experiments, it is more scalable when handling cyclic and dihedral groups $O(q)$ than the symmetric group $O(q^2\text{log}q)$.
>
> __Q1: Search over a subgroup of symmetry.__
>
> Yes, getting the full symmetry group of an ILP is rather difficult, and the symmetry groups considered in our implementations are actually already subgroups of the full one.
>
> However, it is still interesting to investigate the effects of considering subgroups with different sizes. We are now conducting some experiments on it and will submit another comment box to report these results once finished.
>
> __Q2: Symmetry related.__
>
> 1. How to check symmetry
>     - Symmetry groups of ILPs can be efficiently detected by well-developed tools, such as Nauty [2] and Bliss [3].
>
> 2. Non-symmetry problems?
>     - Yes, our method works for non-symmetry problems, as it turns into a "symmetry-agnostic" method trained via classical supervised learning. An example (WA) is given in the response to Weakness 2.
>
> __Q3: Should one apply the algorithm for symmetric problems where the number and the size of similar solutions are unknown?__
>
> As mentioned in Question 2, if the symmetries of the ILP problems are unknown, we should identify them first. Otherwise, our method can not utilize the symmetry information and would turn into a "symmetry-agnostic" one.
>
> __Q4: sub-problems over symemtry groups__
>
> The optimization over symmetry groups requires customization, i.e., we need to design appropriate ways to optimize a specific symmetry group.
> - For cyclic and dihedral groups, it is trivial, since there are only $q$ and $2q$ ($q<n$) possible permutations, respectively.
> - We designed a specific sub-problem to get the optimal permutation from the symmetric group's $q!$ possible permutations. We supplemented a detailed example in Part III of the global author rebuttal, please refer to it.
>
> [1] Gasse, Maxime, et al. "The machine learning for combinatorial optimization competition (ml4co): Results and insights."
>
> [2] McKay B D. Nauty user’s guide (version 2.4).
>
> [3] Junttila T, Kaski P. Conflict propagation and component recursion for canonical labeling

---

> > ### Comment · Reviewer_cWBm · 2024-08-12
> >
> > Thanks for your responses. I will remain the score.

---

### Official Review · Reviewer_HH73 · 2024-07-12

**Soundness:** 4
**Presentation:** 4
**Contribution:** 3
**Rating:** 7
**Confidence:** 4

**Summary:**

This paper aims to enhance solution prediction methods for ILPs that contain certain types of symmetry. The main methodological contribution of the paper is a loss function that considers symmetry, along with an optimization algorithm for it. In particular, the loss function applies a permutation to each optimal solution label, and these permutations are learned along with the solution predictor. This is done via an alternating algorithm that alternates between optimizing (graph) neural network weights with a classical loss function and optimizing this permutation. For symmetric groups where any permutation is allowed, the authors show that the latter problem can be efficiently solved via LP. This method is then combined with three downstream methods that use a solution predictor and tested on four sets of problems with symmetry, showing consistent improvements in obtaining better solutions in all cases.

**Strengths:**

This method overall is a solid and clean approach to incorporate symmetry information in solution prediction methods in ILP. While the alternating algorithm seems quite natural in retrospect, it is a suitable approach particularly with the nice observation that optimizing the permutation for the symmetric group case can be done efficiently via LP. It is also not too complicated to implement. If one is solving a problem with known symmetry via a solution prediction approach, I see little drawback to using this method. The experimental setup is sound and sufficiently extensive, covering four sets of instances and three different methods to integrate this with. Even before reading the computational section, I could see that some computational improvement was expected since it allows the neural network model to not worry about permuted optimal solutions, and the computational section does confirm that with substantial improvements across the board. The paper is overall clear and easy to follow.

**Weaknesses:**

I do not see any major weaknesses in this paper. Perhaps a concern in terms of relevance is that work does apply to a rather narrow class of instances (those with the types of symmetry in the paper) and it also requires prior knowledge of which type of symmetry the problem has. However, this does include a number of important problems in discrete optimization, so I do not view this as a significant issue. Other than that, there are a few parts of the experimental setup that need to be clarified as discussed below, but they can be easily fixed.

**Questions:**

1. It would be helpful for the reader to know earlier in the Introduction that the predicted optimal solution will be coupled with downstreams approaches rather than used directly. Options are to mention this in the introduction, or alternatively move Figure 2 (which is very helpful) near the introduction, as it easier to go through the paper with the big picture in your head. If you do opt to move Figure 2 earlier, consider adding some context in the caption since the reader does not have much context at that point.

2. Please add a description of the problems in the Appendix, preferrably (but not necessarily) with the MILP formulation used, and point out where the symmetry comes from. In addition, indicate how the perturbation for PESPlib was done.

3. I tried to check the anonymous code repository, but it was expired.

4. Could you add standard deviations / errors to Table 2?

5. The Appendix says that the evaluation machine has a GPU, but I cannot clearly find if the training time reported in Table 2 is for GPUs. The reason why this matters is that this paper shows that the LP is not a bottleneck, but the picture might be different if Table 2 were CPU times. In particular, I am assuming that the $r_s$ time includes: GPU training time + CPU LP solves + communication overhead between GPU and CPU. Could you clarify this in the paper?

6. The paper does not describe what is "Tuned CPLEX". In particular, an important question is, do you tune the symmetry breaking parameters in CPLEX? If you do not, would be able to include it in the paper? Please also mention in the paper how the tuning is done overall, and you might as well mention that CPLEX contains its own symmetry handling methods based on classical approaches. Another important set of parameters to tune are primal focus (e.g. heuristics), but I assume that is already included in your tuning.

7. Typos: Line 269: Is $\hat{x}$ supposed to be $\hat{y}$? Line 420 has a typo: "don not".

**Limitations:**

The paper properly indicates the main limitations of this work in Section 7.

---

> ### Author Rebuttal · Authors · 2024-08-07
>
> Thank you for your detailed review and valuable comments. We address each question below.
>
> ___Question 1: suggestion of presenting downstream approaches earlier___
>
> We appreciate your valuable suggestion and already added a short paragraph in the Introduction to explain the downstream part. Meanwhile, Figure 2 is moved to the earlier page adding the necessary explanation in the caption.
>
> Here is the added description: Due to the difficulty of satisfying complicated constraints in ILPs, existing methods are usually equipped with a post-processing module to identify high-quality solutions based on the initial solution predicted by GNNs. A number of methods utilize ILP solvers in the post-processing module. The initial prediction from GNNs is taken as guidance for the ILP solver to solve the target ILP. We call these post-processings as downstream approaches according to how they utilize the prediction as guidance. In our method, we follow their routines and incorporate some downstream approaches.
>
>
> ___Question 2.1: Detailed information for used MILP and symmetry___
>
> Thanks for your valuable suggestion, we have added detailed descriptions of all benchmark problems with their formulations, as well as symmetry groups corresponding to their decision variables. An example is shown in part III of the global author rebuttal. Please refer to it.
>
> ___Question 2.2: indicate how the perturbation for PESPlib was done.___
>
> Problems in PESPlib involve determining optimal schedules for a set of events that occur repeatedly over a fixed period, such as departures of trains and buses. Each problem has a set of events $\mathcal{E}$ and a set of activities $\mathcal{A} \subseteq \mathcal{E} \times \mathcal{E}$ connecting events with each other. Each activity has a weight $w_a$. The goal is to assign an appropriate time $t_i$ to each event $i\in \mathcal{E}$ to meet some certain constraints while minimizing the total time slack weighted by $\{w_a, a\in \mathcal{A}\}$. These weights heavily impact the time assignment. We perturb these weights by introducing Gaussian noises, i.e., $w_a' = w_a + n_a$, where $n_a \sim \mathcal{N}(\mu=w_a, \sigma=0.1*w_a)$.
>
> ___Question 3: I tried to check the anonymous code repository, but it was expired.___
>
> Thanks for catching this. We have fixed this issue.
>
> ___Question 4: Could you add standard deviations/errors to Table 2___
>
> Yes, the results with standard deviations are shown in the brackets as follows:
>
> | Dataset |   |CPLEX|   || Fix&optimize |||| Local Branching |||| Node Selection |||
> |:-:|:-:|:-:|:-:|:-:|:-:|:-:|:-:|:-:|:-:|:-:|:-:|:-:|:-:|:-:|:-:|
> |||||ND|SymILP|gain ||PS|SymILO|  gain |   |MIP-GNN|SymILO|  gain ||
> |IP||0.188(0.15) ||  0.201(0.18) |  0.124(0.10) | 38.4% ||  0.168(0.15) |   0.102(0.09)  | 39.4% |   |0.312(0.30) |   0.190(0.16)  | 39.2% |   |
> |SMSP|| 0.190(0.001) || 0.300(0.002) | 0.180(0.001)| 40.0% || 0.230(0.001) |0.160(0.001)| 30.4% || 1.180(0.004) |0.740(0.004)| 37.3% ||
> |PESP  ||  0.056(0.08) ||0.084(0.14) |0.050(0.06) | 39.8% ||  0.306(0.63) |0.000(0.00)|100% ||  1.899(1.41) |0.280(0.49)  | 85.3% ||
> |  PESPD  ||  3.194(1.94) ||  2.389(1.28) |  0.404(0.31) | 83.1% ||  3.442(2.09) |   0.127(0.14)  | 96.3% ||  3.755(1.88) |3.006(1.65)  |  20%  |   |
>
> Note that the standard deviations among datasets IP, PESP, and PESPD are relatively large because:
> - the reported primal gaps are averaged across instances, while different instances may have very distinct objective values.
> - these three datasets include instances with very different sizes, please refer to Appendix F.1 for more details.
>
>
> ___Question 5: GPU and CPU time___
>
> We are sorry for the confusion, here we add more explanation for Table 2.
> - The time costs reported in lines $r$ and $r_s$ are training time averaged over iterations (update steps). Those in line $r$ are "GPU training time", while those in line $r_s$ are "GPU training time + CPU LP solves + communication overhead between GPU and CPU".
> - The time cost reported in line $t$ is the time of solving LP averaged over instances, they are pure CPU time for LP solving.
> - The batch size is set to $B = 16$, so in each iteration, the number of LPs is 16.
> - Approximately, $t*B$ (CPU time) + $r$ (GPU time) $\approx$ $r_s$ (CPU+GPU+communication time). Take the column of "IP" as an example, 5.54 + 0.029 * 16 = 6.004 $\approx$ 6.01. The communication time cost is quite small.
>
> ___Question 6.1: Are symmetry-related parameters of CPLEX tuned in experiments?___
>
> Yes, we tuned two CPLEX hyper-parameters: "emphasis switch" (which balances speed, feasibility, optimality, and moving bounds, etc.) and "symmetry breaking" (the level of symmetry-breaking)
>
>
> ___Question 6.2: How CPLEX is tuned.___
>
> The tuning is conducted through a grid search strategy on the validation set for each dataset. We selected the set of hyper-parameters that produced the best average primal gap within a time limit of 800 seconds. These hyper-parameters are then used for evaluation on the test set. Different datasets can have distinct sets of tuned hyper-parameters.
>
> Indeed, as pointed out by [1], commercial solvers such as CPLEX have their own symmetry-handling methods including orbital fixing.
>
>
> ___Question 6.3: Primal focus___
>
> The setting of the primal focus is included in the "emphasis switch" hyper-parameter, which was considered in our experiments. It has 6 choices: balanced, feasibility, optimality, bestbound, hiddenfeas, and heuristic. The "heuristic" option emphasizes finding high-quality feasible solutions earlier.
>
>
>
> ___Question 7: Typos: Line 269: Is $\hat{x}$ supposed to be $\hat{y}$? Line 420 has a typo: "don not".___
>
> Yes, thank you for the correction, we have fixed them in the revised manuscript.
>
>
> [1] Pfetsch, M.E. and Rehn, T., 2019. A computational comparison of symmetry handling methods for mixed integer programs. Mathematical Programming Computation, 11, pp.37-93.

---

> > ### Comment · Reviewer_HH73 · 2024-08-11
> >
> > Thank you for the response. All my questions have been addressed and I will maintain my rating.

---

### Official Review · Reviewer_7rcN · 2024-07-12

**Soundness:** 3
**Presentation:** 3
**Contribution:** 3
**Rating:** 6
**Confidence:** 4

**Summary:**

The paper provides an ML-based framework, SymLo, for solving MILPs that leverages symmetries of ILPs to improve ML performance. SymLo takes into account symmetric groups on the solutions and formulates a learning task with respect to the model parameters and the permutations that belong to the symmetric group of each instance. The loss is challenging to minimize, and an alternating algorithm is proposed to mitigate the issues. In experiments, the method is evaluated on three different tasks and on four benchmarks.

**Strengths:**

1. The proposed formulation and method for symmetry-aware learning for ILPs is novel.

2. Empirical results are promising, suggesting its usefulness for instances that have symmetric properties in the solutions. It is also applicable to different ILP learning tasks.

3. The paper is engaging and well-written.

**Weaknesses:**

This is the second time I have reviewed this paper. I am satisfied with the authors in addressing my previous comments.

Nonetheless, I still believe that evaluating the method on a third task would make the contribution of this work more solid and convincing.  The fix&optimize task and the local branching task are similar to each other. I am not surprised to see that it works on one if it already has worked for the other. Some tasks that you could consider: Initial Basis Selection [1][2], Backdoor variable predictions [3]

[1] Fan, Zhenan, et al. "Smart initial basis selection for linear programs." International Conference on Machine Learning. PMLR, 2023.

[2] Zhang, Yahong, et al. "MILP-FBGen: LP/MILP Instance Generation with Feasibility/Boundedness." Forty-first International Conference on Machine Learning.

[3] Ferber, Aaron, et al. "Learning pseudo-backdoors for mixed integer programs." International Conference on Integration of Constraint Programming, Artificial Intelligence, and Operations Research. Cham: Springer International Publishing, 2022.

2. The limitation of generalizing the method to larger instances was discussed with the other reviewers last time. It would be important to include such a discussion in the main paper to provide a more comprehensive understanding of the method's pros and cons.

**Questions:**

I don't have specific questions.

**Limitations:**

The authors talk about some limitations of the work, but not all that are known.

---

> ### Author Rebuttal · Authors · 2024-08-07
>
> ___Weakness 1: evaluating the method on an additional task___
>
> Thank you very much for listing potential tasks that we can consider in our experiments. Regarding the relevance, we have cited them in our revised manuscript.
>
> We agree that evaluating on a third task could improve our work's contribution, and we are now conducting experiments on some potential methods. Since the training and evaluation do require a certain time, we are not sure whether complete results could be obtained within the rebuttal period. If done, we will submit a follow-up comment to present these results.
>
>
>
> ___Weakness 2: generalization to larger instances___
>
> Thank you again for your valuable comments. We will add such a discussion in our main paper. Similar to Weakness 1, we are now conducting experiments on larger instances and will report corresponding results once finished.

---

> > ### Comment · Reviewer_7rcN · 2024-08-12
> >
> > Regarding weakness 1, I gave you the same comment last time during ICML review. You would have already addressed it if you wanted to.
> >
> > Nevertheless, I will keep my score.

---

### Author Rebuttal · Authors · 2024-08-07

Thanks all reviewers for their detailed review and valuable comments. In the global rebuttal, we would like to supplement some common clarifications.

___Part I. Clarifications on some notions___

---
**1. Permutation :** A permutation is a bijective mapping from a set $I^q=\\{1,\\dots,q\\}$ onto iteself, e.g., an identity permutation on $I^3$ is (1&rarr;1,2&rarr;2, 3&rarr;3) and a reverse permutation is (1&rarr;3,2&rarr;2, 3&rarr;1). For simplicity, **we omit the ordered and identity preimage part in the following text**, e.g., (1,2,3) and (3,2,1) denote the identity and reverse permutations, respectively. A permutation $\\pi$ acts on a vector $x=[x_1,\\dots,x_q]^\\top$ by rearranging its elements, i.e., $\\pi(x)=[x_{\\pi(1)},\\dots,x_{\\pi(q)}]^\\top$

**2. Permutation group :** A permutation group $G$ is a group [1] whose (i) **elements** are permutations, (ii) and group **operation** $\\circ$ is _combination_, i.e. $a\\circ b=a(b),\\forall a,b\\in G$.

**3. Symmetric group, cyclic group, and dihedral group:** These three groups are typical permutation groups [1]. Below we list examples for each one on set $I^3$.
- Symmetric group: $S_3=\\{(1,2,3),(1,3,2),(2,1,3),(2,3,1),(3,1,2),(3,2,1)\\}$ (**all permutations**)
- Cyclic group: $C_3=\\{(1,2,3),(3,1,2),(2,3,1)\\}$ (**rotation**)
- Dihedral group: $D_3=C_3\\cup\\{(3,2,1),(2,1,3),(1,3,2)\\}$ (**rotation + reflection**)

**4. Symmetry group:** A symmetry group is a permutation group associated with specific ILPs. It is used to express an ILP's intrinsic symmetry. For any feasible solution $x$, each permutation in this group can map $x$ to another feasible solution with the same objective.

[1] Grayland A. Automated static symmetry breaking in constraint satisfaction problems[D].

___Part II. Recall and outline the steps of our method___

---
1. Get a dataset $\\{(s_i,x_i,G_i)\\}_{i=1}^N$ with $N$ samples.
   > - $s_i$ : $i$-th ILP instance
   > - $x_i$ : a feasible solution of $s_i$
   > - $G_i$ : a symmetry group of $s_i$.
2. Training a NN model $f_\\theta$ by $\\arg\\min_{\\theta \\in \\Theta, \\pi_i \\in G_i} \\frac{1}{N}\\sum_{i=1}^N\\ell (f_\\theta(s_i),\\pi_i(x_i))$.
   > - $\\pi_i$ : a **decision variable** for $s_i$, i.e., a permutation to be selected from $G_i$.
   > - $\\ell$ : a loss function
   > - Decisions $\\theta$ and $\\pi_i$ are optimized in an alternating manner.

    2.1. Optimize $\\pi_i$ by solving a sub-problem $\\pi_i^{k+1}=\\arg\\min_{\\pi_i\\in G_i}\\frac{1}{N}\\sum_{i=1}^N\\ell(f_{\\theta^k}(s_i),\\pi_i(x_i))$
    >> - **this sub-problem needs customization for different symmetry groups.**
   >> - **Supplementary details are in part III**

    2.2. Optimize $\\theta$ by $\\theta^{k+1}=\\arg\\min_{\\theta\\in\\Theta}\\frac{1}{N}\\sum_{i=1}^N\\ell(f_\\theta(s_i),\\pi^{k+1}_i(x_i))$

   >> - A trained GNN model after $K$ iterations is $f_{\\theta^K}$.
3. Evaluation on a test instance $s_t$

    3.1. Predict an inital solution by $\hat{x}\_t = f\_{\theta^K}(s_t)$.

    3.2. Post-processing to identify high-quality solutions based on $\\hat{x}_t$


___Part III. Specification of the considered problems___

---
**1. Bin packing problem:**

**Description :** This is the toy example in Appendix B.0.1, it packs $I$ items into $J$ bins, aiming at a minimum number of used bins without exceeding the capacity.

**Formulation :**
\\begin{align}\\min_{x_{ij},y_j\\in\\{0,1\\}}~&y_1+y_2+y_3\\notag\\\\&a_1x_{1j}+a_2x_{2j}+a_3x_{3j}\\leq By_j,&\\forall j\\in J\\\\&x_{i1}+x_{i2}+x_{i3}=1,&\\forall i\\in I\\end{align}
where $y_j=1$ denotes $j$-th bin is used and $x_{ij}=1$ denotes $i$-th item is placed in $j$-th bin.

**Symmetry group :** All bins have the same capacity $B$, which leads to a symmetry of reordering them.

Specifically, let $X=\\begin{bmatrix}y_{1}&y_{2}&y_{3}\\\\
x_{11}&x_{12}&x_{13}\\\\x_{21}&x_{22}&x_{23}\\\\x_{31}&x_{32}&x_{33}\\end{bmatrix}$ be an optimal solution and $X_{:j}$ be its $j$-th column, one can easily check that $[X_{:1},X_{:3},X_{:2}],[X_{:2},X_{:1},X_{:3}],[X_{:2},X_{:3},X_{:1}],[X_{:3},X_{:1},X_{:2}],[X_{:3},X_{:2},X_{:1}]$ are all equivalent to $X$. Formally, it has a **symmetric group** w.r.t. bins $J$.

**2. Item placement (IP) problem:**

**Description :** Similar to bin packing, IP places $I$ items to $J$ bins with $K$ types of resouces. The difference lies in its goal of balancing packed resources on each bin.

**Formulation:**
\\begin{align}&\\underset{x,y,z}{\\text{min}}&&\\sum_{j\\in J}\\sum_{k\\in K}\\alpha_{k}y_{jk}+\\sum_{k\\in K}\\beta_k z_{k}\\notag\\\\&\\text{s.t.}&&\\sum_{j\\in J}x_{ij}=1&&&\\forall i\\in I\\\\&&&\\sum_{i\\in I }a_{ik}x_{ij}\\leq b_{k}&&&\\forall j\\in J,\\forall k\\in K\\\\&&&\\sum_{i\\in I}d_{ik}x_{ij}+y_{jk}\\geq 1&&&\\forall j\\in J,\\forall k\\in K\\\\&&& y_{jk}\\leq z_{k}&&&\\forall j\\in J,\\forall k\\in K\\\\&&& x_{ij}\\in\\left\\{0,1\\right\\}&&&\\forall i\\in I,\\forall j\\in J\\\\&&& y_{jk}\\geq 0&&& \\forall j\\in J,\\forall k\\in K\\end{align}

where $x_{ij}=1$ denotes assigning item $i$ to bin $j$, $a_{ik}$ and $d_{ik}$ are resource coefficients. Besides, $y_{jk}$ and $z_k$ are auxiliary variables tracking resource imbalance.

**Symmetry group :** Each bin $j$ can also be permuted. So it has a **symmetric group $S_{|J|}$** w.r.t. the bins $J$. Specifically, let $X \\in \\{0,1\\}^{|I|\\times |J|}$ be a feasible solution of with its $(i,j)$-th entry as the value of variable $x_{ij}$. Then arbitrary permutation $\\pi \\in S_{|J|}$ acting on its columns $\\{X_{:j}, \\forall j\\in J\\}$ yields an equivalent solution $\\left[X_{:\\pi(1)},X_{:\\pi(2)},\\dots,X_{:\\pi(|J|)} \\right]$.

**Sub-problem over the symmetry group :** Since the permutations of the symmetric group involve all possible reordering of the matrix columns. The sub-problem $\\min_{\\pi \\in S_{|J|}} \\ell(f_\\theta(s),\\pi(X))$ can equivalently modeled as $\\min_{P \\in \\mathcal{P}} \\ell(f_\\theta(s),XP)$, where $\\mathcal{P}$ is all permutation matrices (refer to Section 4.2.1).

---

### Decision · Program_Chairs · 2024-09-25

**Decision:**

Accept (poster)

**Comment:**

This paper proposes a machine learning framework to solve mixed integer linear programming (MLIP). The main idea is to use a symmetry-aware loss function that incorporates permutations of solutions, which allows the model to learn more stable patterns for symmetric solutions. The authors also propose an alternating algorithm to train the neural network and the solution permutations. The framework was tested on three different tasks and four benchmark datasets, showing consistent improvements over traditional, symmetry-agnostic methods.

The reviewers unanimously support acceptance of the paper. I also recommend acceptance for this work.